# Mixed Dynamics In Linear Networks: Unifying the Lazy and Active Regimes

**Zhenfeng Tu**
Courant Institute
New York University
New York, NY 10012
zt2255@nyu.edu

**Santiago Aranguri**
Courant Institute
New York University
New York, NY 10012
aranguri@nyu.edu

**Arthur Jacot**
Courant Institute
New York University
New York, NY 10012
arthur.jacot@nyu.edu

## Abstract

The training dynamics of linear networks are well studied in two distinct setups: the lazy regime and balanced/active regime, depending on the initialization and width of the network. We provide a surprisingly simple unifying formula for the evolution of the learned matrix that contains as special cases both lazy and balanced regimes but also a mixed regime in between the two. In the mixed regime, a part of the network is lazy while the other is balanced. More precisely the network is lazy along singular values that are below a certain threshold and balanced along those that are above the same threshold. At initialization, all singular values are lazy, allowing for the network to align itself with the task, so that later in time, when some of the singular value cross the threshold and become active they will converge rapidly (convergence in the balanced regime is notoriously difficult in the absence of alignment). The mixed regime is the 'best of both worlds': it converges from any random initialization (in contrast to balanced dynamics which require special initialization), and has a low rank bias (absent in the lazy dynamics). This allows us to prove an almost complete phase diagram of training behavior as a function of the variance at initialization and the width, for a MSE training task.

## 1 Introduction

Whether in linear networks or nonlinear ones, there has been a lot of interest in the distinction between the lazy regime [27] and the active regime [16, 43, 15, 52, 13] as the number of neurons grows towards infinity. In the lazy regime the training dynamics become linear, so that they can be easily described in terms of the Neural Tangent Kernel (NTK) [27, 9, 51, 36], while the active regime exhibits complex nonlinear dynamics. While our understanding of the active regime remains much more limited, it appears to be characterized by the emergence of feature learning[22, 52], and of a form of sparsity [3, 11, 2, 1] (the type of sparsity observed depends on the network type [11, 19, 25, 24], but we will focus on fully-connected linear networks which exhibit a rank sparsity in the learned linear map [7, 35, 28, 47]) which are both absent in the lazy regime .

Note that even though it is common to talk of the 'the' active regime, we do not know yet whether there is only one or multiple active regimes. Indeed the term active regime is usually used to describe any regime that differs from the lazy regime and exhibit some form of feature learning. Though we do not have an complete understanding of where the lazy regimes ends and the active regime(s) start, we know that the lazy regime requires extreme overparametrization (a large number of neurons in comparison to the number of datapoints) [5, 20], a 'large' initialization of the weights [15], a small learning rate, and early stopping when using a cross-entropy loss or weight decay . Indeed, active regimes have been observed by breaking either of these requirements: taking limits with mild or no overparametrization [10], taking smaller or even vanishingly small initializations [35, 28], using large

learning rates [33] or SGD [42, 47], or studying the late training dynamics with the cross-entropy loss [30, 17] or weight decay [34, 39, 29, 26]. Though each of these can lead to active regimes with significantly different dynamics, they often lead to similar types of feature learning and sparsity.

In this paper, we study this transition in the context of linear networks and focus mainly on the effects of the width $w$ and the variance of the weights at initialization $\sigma^2$, and give a precise and almost complete phase diagram, showing the transitions between lazy and active regimes. In this setting, we will show that there typically is only 'one' active regime, which is the same (up to approximation) as the already well-studied balanced regime [44, 7, 8].

But our result also paint a more subtle picture than the lazy/active dichotomy. We propose a more granular approach, where at a certain time some part of the network can be in the lazy regime, while others are in the active or balanced regime. More precisely the network is lazy along the singular values of the matrix represented by the network that are smaller than $\sigma^2 w$, and in the active regime along the singular values larger than $\sigma^2 w$.

## 1.1 Contributions

We consider the training dynamics of shallow linear networks $A_\theta = W_2 W_1$ and show that for large enough width $w$ (the inner dimension), and a iid $\mathcal{N}(0, \sigma^2)$ initialization of all weights, the dynamics of $A_{\theta(t)}$ as a result of training the parameters $\theta = (W_1, W_2)$ with GD/GF on the loss $\mathcal{L}(\theta) = C(A_\theta)$ for a general matrix cost $C$ with learning rate $\eta$ is approximately given by the self-consistent dynamics

$$\partial_t A_{\theta(t)} \approx -\eta \sqrt{A_\theta A_\theta^T + \sigma^4 w^2 I} \nabla C(A_\theta) - \eta \nabla C(A_\theta) \sqrt{A_\theta^T A_\theta + \sigma^4 w^2 I}. \tag{1}$$

These dynamics contain as special cases both the lazy dynamics

$$\partial_t A_{\theta(t)} \approx -2\eta \sigma^2 w \nabla C(A_\theta)$$

when $\sigma^2 w \gg \lambda_{max}(A_\theta)$ and the balanced dynamics

$$\partial_t A_{\theta(t)} = -\eta \sqrt{A_\theta A_\theta^T} \nabla C(A_\theta) - \eta \nabla C(A_\theta) \sqrt{A_\theta^T A_\theta}$$

when $\sigma^2 w \ll \lambda_{min}(A_\theta)$. But it also reveals the whole spectrum of mixed dynamics in between, where some singular values of $A_\theta$ are below the $\sigma^2 w$ threshold and some are above it.

This suggests that the lazy/active transition is best understood at a more granular level, where at each time $t$ every singular value of $A_\theta$ can either be lazy or active/balanced. The mixed regime is the best of both worlds: on one hand, since $\sqrt{A_\theta A_\theta^T + \sigma^4 w^2 I}$ is always positive definite, the network can never get stuck at a saddle/local minimum as can happen in the balanced regime, on the other hand there is a momentum effect where the dynamics along large singular values is much faster than along the small ones, leading to an incremental learning behavior and a low-rank bias, which is absent in lazy learning. By choosing the threshold $\sigma^2 w$ adequately, one can best take advantage of these two phenomenon.

Finally, we focus on the task of recovering a low-rank $d \times d$ matrix $A^*$ from noisy observations $A^* + E$, training on the MSE error $\frac{1}{d^2} \|A_\theta - (A^* + E)\|_F^2$ in the limit as the dimension $d$, width $w$ and variance $\sigma^2$ scale together with scaling laws $w = d^{\gamma_w}$ and $\sigma^2 = d^{\gamma_{\sigma^2}}$. We describe the training dynamics for almost all reasonable scalings $\gamma_w, \gamma_{\sigma^2}$ leading to a phase diagram with two main regimes:

- **Lazy** ($1 < \gamma_{\sigma^2} + \gamma_w$) where all singular values remain below the threshold $\sigma^2 w$ throughout training, and where the network fails to recover $A^*$ due to the absence of low-rank bias.

- **Active** ($1 > \gamma_{\sigma^2} + \gamma_w$) where $K = \text{Rank} A^*$ singular values pass the threshold and fit $A^*$ before the other singular values have time to fit the noise $E$, leading to the recovery of $A^*$.

There are two other degenerate regimes that we avoid: the underparametrized regime when $w < d$ (or $\gamma_w \ll 1$) where the rank is constrained by the network architecture rather than the training dynamics, and the noisy regime $2\gamma_{\sigma^2} + \gamma_w + 1 > 0$ where the variance of the entries of $A_{\theta(0)}$ at initialization is infinite.

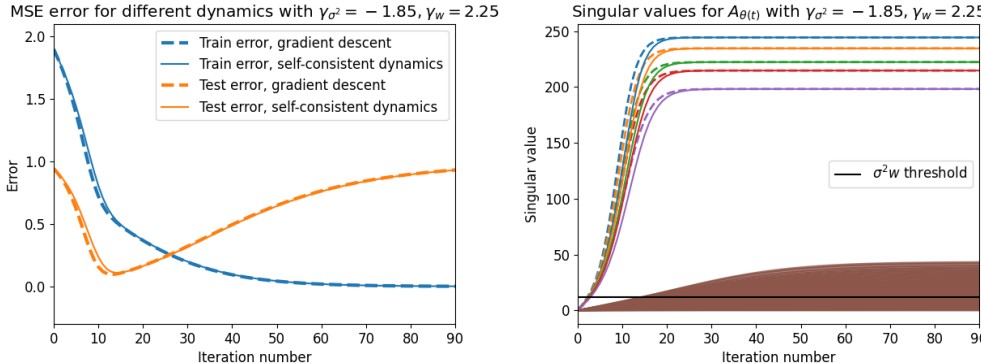

Figure 1: For both plots, we train either using gradient descent or the self-consistent dynamics from equation (1), with the scaling $\gamma_{\sigma^2} = -1.85$, $\gamma_w = 2.25$ which lies in the active regime. (Left panel): We plot train and test error for both dynamics. We observe that the train/test error for gradient descent is very close to the train/test error for the self-consistent dynamics. (Right panel): We plot with a solid line the singular values of $A_{\theta(t)}$ when running the self-consistent dynamics, and use a dashed line for the singular values from running gradient descent. In this experiment, $\text{Rank}A^\star = 5$. We use different colors for the 5 largest singular values and the same color for the remaining singular values. We can see how the 5 largest singular values 'speed up' as they cross the $\sigma^2 w$ threshold, allowing them to converge earlier than the rest. The minimal test error is achieved in the short period where the large singular values have converged but not the rest.

## 1.2 Previous Works

Linear networks have been used as a testing ground, a stepping stone on the way to understand nonlinear networks. Linear networks and their training dynamics are in many ways much simpler than nonlinear ones, but in spite of a long research history, our understanding remains limited.

The setting that is best understood is that of diagonal linear networks where the dynamics decouple along the diagonal entries leading to an incremental learning behavior and a sparsity bias [44, 3, 45, 23, 41], some of this analysis has been extended to include effects of initialization scale [48] and SGD [42]. While the same decoupling happens in general linear with diagonal initializations and diagonal task, it remains an extremely strong assumption.

Some work has been done to prove similar incremental learning dynamics outside the diagonal case [35, 28, 31] where the incremental aspect can be understood as the parameters going from saddle to saddle. For shallow linear networks, the training dynamics with MSE can be explicitly solved [21] but remain very complex so that one needs to assume some form of alignment to guarantee convergence [14]. For deeper networks there exists explicit formulas in the mean-field limit where the number of neurons grows to infinity [18], these results can of course be applied to the special case of shallow nets, our paper goes further by giving self-consistent dynamics for the full matrix, revealing the lazy/active transition, and also extends the analysis to finite widths.

A very powerful tool in the analysis of a linear network is its training invariants, and the balancedness condition which greatly simplifies the dynamics [6, 7]. Balanced networks exhibit a momentum effect, where the training dynamics along a singular value $s_i$ have 'speed' proportional to $s_i$ itself (or $s_i$ to some power), while this momentum effect seems to be key to understand the low-rank bias of linear networks [8], it also means that one needs to guarantee that the dynamics never approach zero, which is one the main hurdle towards proving convergence in balanced networks. To solve this issue, recent work has focused on initialization that slightly imbalanced [49, 37, 46, 38, 50]. This suggests that it is key to find the right balance between balancedness and imbalancedness to obtain both fast convergence and low-rank bias.

In a concurrent work [32] a similar transition between lazy and active regimes is observed, and the same mixed dynamics are derived for a specific initialization. In contrast, we prove that these dynamics are approximately true with high probability for random Gaussian initializations, which is the standard initialization scheme for neural networks.

## 1.3  Setup

We will study shallow linear networks (or matrix factorization) where a $d_{out} \times d_{in}$ matrix $A_\theta$ is represented as the product of two matrices $A_\theta = W_2 W_1$, where the weight matrices $W_1$ and $W_2$ are respectively $w \times d_{in}$ and $d_{out} \times w$ dimensional, for some width $w$. The parameters $\theta$ of the network are the concatenation of the entries of both submatrices $\theta = (W_1, W_2)$.

The parameters $\theta$ are learned in the following manner: they are initialized as i.i.d. Gaussian $\mathcal{N}(0, \sigma^2)$, and then optimized with gradient descent to minimize a loss $\mathcal{L}(\theta) = C(A_\theta)$. Though most of our analysis works for general convex costs $C : \mathbb{R}^{d_{out} \times d_{in}} \to \mathbb{R}$ on matrices, we will in the second part focus on the task of recovering a low-rank matrix $A^*$ from noisy observations $A^* + E$, by training a linear network $A_\theta$ on the MSE loss

$$\mathcal{L}(\theta) = \frac{1}{d^2} \|A_\theta - (A^* + E)\|_F^2 \,.$$

The width $w$ allows us to control the over parametrzation, indeed the set of matrices that can be represented by a network of width $w$ is the set $\mathcal{M}_{\leq w}$ of matrices of rank $w$ or less. The overparametrized regime is when $w \geq \min\{d_{in}, d_{out}\}$ because all matrices can be represented in this case.

## 1.4  Lazy Dynamics

The evolution of the weight matrices during gradient descent with learning rate $\eta$ is given by

$$W_1(t+1) = W_1(t) - \eta W_2^T(t) \nabla C(A_{\theta(t)})$$
$$W_2(t+1) = W_2(t) - \eta \nabla C(A_{\theta(t)}) W_1^T(t)$$

where we view the gradient $\nabla C(A_{\theta(t)})$ of the cost $C$ as a $d_{out} \times d_{in}$ matrix, which for the MSE cost equals $\nabla C(A_{\theta(t)}) = 2d^{-2}(A_{\theta(t)} - (A^* + E))$.

But we care more about the evolution of the complete matrix $A_{\theta(t)} = W_2(t)W_1(t)$ induced by the evolution of $W_1(t), W_2(t)$, which can be approximated by

$$A_{\theta(t+1)} = A_{\theta(t)} - \eta W_2(t) W_2^T(t) \nabla C(A_{\theta(t)}) - \eta \nabla C(A_{\theta(t)}) W_1^T(t) W_1(t) + O(\eta^2). \qquad (2)$$

Thus we see that if we can describe the matrices $C_1 = W_1^T W_1$ and $C_2 = W_2 W_2^T$ throughout training, then we can describe the evolution of $A_{\theta(t)}$.

When $w$ is very large, we end up in the lazy regime where the parameters move enough up to a time $t$ to change $A_{\theta(t)}$, but not enough to change $C_1, C_2$[1], allowing us to make the approximation $C_i(t) \approx C_i(0)$. Furthermore at initialization these matrices concentrate as $w \to \infty$ around their expectations $\mathbb{E}[C_1] = \sigma^2 w I_{d_{in}}$, $\mathbb{E}[C_2] = \sigma^2 w I_{d_{out}}$. The GD dynamics can then be approximated by the much simpler dynamics:

$$A_{\theta(t+1)} = A_{\theta(t)} - 2\eta \sigma^2 w \nabla C(A_{\theta(t)}),$$

which are equivalent to doing GD on the cost $C$ directly with a learning rate of $2\eta \sigma^2 w$.

One can then easily prove exponential convergence for any convex cost $C$ following the convergence analysis of traditional linear models. But we can see the absence of feature learning from the fact that the covariance $C_1$ of the 'feature map' $W_1$ is (approximately) constant. More problematic in the context of low-rank matrix recovery is the absence of low-rank bias, indeed one can easily solve the dynamics to obtain

$$A_{\theta(t)} = (A^* + E) + (1 - 4d^{-2}\eta\sigma^2 w)^t (A_{\theta(0)} - (A^* + E)),$$

and since $\mathbb{E}A_{\theta(0)} = 0$ we obtain

$$\mathbb{E}\left[A_{\theta(t)}\right] = \left(1 - (1 - 4d^{-2}\eta\sigma^2 w)^t\right)(A^* + E).$$

---

[1]To be more precise the direction in parameter space that change $C_1, C_2$ are approximately orthogonal to those that change $A_\theta$, and GD/GF only moves along the later direction.

The expected test error $\mathbb{E} \left\| A_{\theta(t)} - A^* \right\|^2$ is therefore lower bounded by

$$\left\| \mathbb{E} A_{\theta(t)} - A^* \right\|^2 = \left\| (1 - 4d^{-2}\eta\sigma^2 w)^t A^* + (1 - (1 - 4d^{-2}\eta\sigma^2 w)^t) E \right\|^2$$

which never approaches zero.

In linear networks, there is no advantage to being in the lazy regime, as we simply recover a simple linear model at an additional cost of more parameters and thus more compute. But we will see that a short period of lazy regime at the beginning of training plays a crucial role in making sure that the subsequent active regime starts from an 'aligned' state.

## 1.5   Balanced Dynamics

There has been much more focus on so-called balanced linear networks, which are networks that satisfy the balanced condition $W_1 W_1^T = W_2^T W_2$. If the network is balanced at initialization, it remains so throughout training, because, the difference $W_1 W_1^T - W_2^T W_2$ is an invariant of GF (and an approximate invariant of GD with small enough learning rate).

First observe that the balanced condition implies the following shared eigendecomposition $W_1 W_1^T = W_2^T W_2 = USU^T$. This implies the following shared SVD decompositions $W_1 = U\sqrt{S}U_{in}^T$, $W_2 = U_{out}\sqrt{S}U^T$ and $A_\theta = U_{out}SU_{in}^T$. Furthermore, we have $C_1 = U_{in}SU_{in}^T = \sqrt{A_\theta^T A_\theta}$ and $C_2 = U_{out}SU_{out}^T = \sqrt{A_\theta A_\theta^T}$, which leads to self-consistent dynamics for $A_{\theta(t)}$:

$$A_{\theta(t+1)} = A_{\theta(t)} - \eta\sqrt{A_{\theta(t)}A_{\theta(t)}^T}\nabla C(A_{\theta(t)}) - \eta\nabla C(A_{\theta(t)})\sqrt{A_{\theta(t)}^T A_{\theta(t)}} + O(\eta^2).$$

Now these dynamics are quite complex in general, and it remains difficult to prove convergence. Indeed one can easily find initializations $A_{\theta(0)}$ that will not converge, for example if $A_{\theta(0)} = 0$ then GD will remain stuck there. A lot of work has been dedicated to finding conditions that guarantee the convergence of the above dynamics [6, 14], but these assumptions are often quite strong.

The simplest initialization that guarantees convergence (and the one that will be most relevant to our analysis) is the positively aligned initialization. If at initialization $A_{\theta(0)}$ and $A^* + E$ are 'aligned', i.e. shares the same singular vectors $A_{\theta(0)} = U_{out}SU_{in}^T$ and $A^* + E = U_{out}S^*U_{in}^T$, then they will remain aligned throughout training $A_{\theta(t)} = U_{out}S(t)U_{in}^T$ and the dynamics decouple along each singular value

$$s_i(t + 1) = s_i(t) + 2\eta \left| s_i(t) \right| (s_i^* - s_i(t)) + O(\eta^2).$$

Since we always have $s_i^* \geq 0$, then for small enough learning rates $\eta$, we see that if $s_i(0) \in (0, s_i^*]$ it will grow monotonically and converge to $s_i^*$; if $s_i(0) > s_i^*$ it will decrease monotonically to $s_i^*$, and if $s_i(0) \leq 0$ it will increase and converge to 0. Thus one can guarantee convergence if we further assume positive alignment $s_i(0) > 0$.

The advantage is that there is a momentum effect in the form of the prefactor $|s_i(t)|$, which implies that the dynamics along large singular values are faster than along small ones. As a result, if all singular values are initialized with the same small value, then they will at first grow very slowly until they reach a critical size where the momentum effect will make them converge very fast. The singular values aligned with the top singular values of $A^* + E$ will reach this threshold much faster, and they will therefore converge to approximately their final value $s_i = s_i^*$ at a time when the other singular values are still basically zero. If we stop training at this time then the linear network will have essentially learned only the top $K$ singular values of $A^* + E$, which is a good approximation for $A^*$, leading to a small test error (see [23] for details).

But this analysis relies on the very strong assumption of positive alignment at initialization. If we do not assume a positive alignment and assume that the $s_i$ are random (i.i.d. w.r.t. a symmetric distribution), then each $s_i$ has probability $1/2$ of starting with a negative alignment and getting stuck at zero, which means that with high probability training will fail to recover $A^*$ and will recover only a random subset of the singular values of $A^*$. The presence of these attractive saddles shows the complexity of the balanced dynamics.

A limitation of this approach is that it requires a quadratic cost and a very specific initialization, and in the case of positive alignment, an initialization that requires knowledge of the (SVD of the)

true function $A^*$. Nevertheless, the positively aligned and balanced dynamics seem to capture some qualitative phenomenon that has been observed empirically outside of this restricted setting. This is the phenomenon of incremental learning, where if the singular values are initialized as very small, they first grow very slowly, but the multiplicative momentum will lead to come up one by one in a very abrupt manner, and this leads to a low rank bias where the network first only fits the largest singular value, then two largest, and so on. More generally, this can be interpreted as the network performing a greedy low-rank algorithm [35].

Our analysis will confirm the fact that positive alignment happens naturally as a result of a short period of lazy training, allowing us to prove similar decoupling and incremental learning for a general random initialization.

*Remark.* We can define the time dependent map $\Theta(G;t) = C_2(t)G + GC_1(t)$, so that the GD dynamics can be rewritten as $A_{\theta(t+1)} = A_{\theta(t)} - \eta\Theta(\nabla C(A_{\theta(t)}), t) + O(\eta^2)$. The map $\Theta$ is none other than the NTK for shallow linear networks, but it has also been called the preconditioning matrix in previous work [7]. The lazy regime is then characterized by the NTK $\Theta$ being approximately equal to the time-independent NTK $\Theta^{\text{lazy}}(G) = 2\sigma^2 wG$, whereas the balanced regime is characterized by the time-dependent $\Theta^{\text{bal}}(G;t) = \sqrt{A_{\theta(t)}A_{\theta(t)}^T}G + G\sqrt{A_{\theta(t)}^T A_{\theta(t)}}$, with the distinction that the time dependence is only through $A_{\theta(t)}$.

## 2 Mixed Lazy/Balanced Dynamics

Both lazy and balanced dynamics have the surprising but very useful property that the evolution of the network matrix $A_\theta$ is approximately self-consistent: the evolution of $A_\theta$ can be expressed in terms of itself. The lazy approximation becomes correct for a sufficiently large initialization, while the balanced one is correct for a balanced initialization. However, for most initializations, neither of these approximations are correct.

We fill this gap by providing a self-consistent evolution of $A_\theta$ that applies for any initialization scale:

$$\partial_t A_{\theta(t+1)} \approx -\eta\sqrt{A_{\theta(t)}A_{\theta(t)}^T + \sigma^4 w^2 I}\nabla C(A_t) - \eta\nabla C(A_t)\sqrt{A_{\theta(t)}^T A_{\theta(t)} + \sigma^4 w^2 I}.$$

This approximation is formalized in the following theorem, denoting $\hat{C}_1(t) = \sqrt{A_{\theta(t)}^T A_{\theta(t)} + \sigma^4 w^2 I}$ and $\hat{C}_2(t) = \sqrt{A_{\theta(t)}A_{\theta(t)}^T + \sigma^4 w^2 I}$

**Theorem 1.** *For a linear net $A_\theta = W_2 W_1$ with width $w$, initialized with i.i.d. $\mathcal{N}(0, \sigma^2)$ weights and trained with Gradient Flow, we have with high probability that for all time $t$,*

$$\left\|C_1(t) - \hat{C}_1(t)\right\|_{op}, \left\|C_2(t) - \hat{C}_2(t)\right\|_{op} \leq min\left\{O(\sigma^2 w), O\left(\sqrt{\frac{d}{w}}\|C_1(t)\|_{op}\right)\right\}.$$

*Proof.* (sketch) The quantity $W_1 W_1^T - W_2^T W_2$ is invariant under GF (and approximately so under GD) and it is approximately equal to $\sigma^2 w(P_1 - P_2)$ for two orthogonal projections $P_1, P_2$ (at initialization and for all subsequent times because of the invariance). We therefore have

$$W_1^T(W_1 W_1^T - W_2^T W_2)^2 W_1 \approx \sigma^4 w^2 W_1^T(P_1 + P_2)W_1 \approx \sigma^4 w^2 C_1.$$

Thus the pairs $C_1, C_2$ approximately satisfy the following equations:

$$0 \approx C_1^3 - A_\theta^T A_\theta C_1 - C_1 A_\theta^T A_\theta - \sigma^4 w^2 C_1 + A_\theta^T C_2 A_\theta$$
$$0 \approx C_2^3 - A_\theta A_\theta^T C_2 - C_2 A_\theta A_\theta^T - \sigma^4 w^2 C_2 + A_\theta C_1 A_\theta^T.$$

The pair $\hat{C}_1, \hat{C}_2$ is a solution of the above, and one can show that $C_1, C_2$ must approach them and not any of the other solutions. $\qed$

The takeaway from theorem 1 is the following.

1. In the lazy regime where $\|C_1(t)\|_{op} + \|C_2(t)\|_{op} \leq O(\sigma^2 w)$, then $\|C_1(t) - \hat{C}_1(t)\|_{op} \leq \sqrt{d/w}\|C_1(t)\|_{op} << \|C_1(t)\|_{op}$.

2. In the active regime where $\|C_1(t)\|_{op}/\sigma^2 w > d^\varepsilon >> 1$, then $\|C_1(t) - \hat{C}_1(t)\|_{op} \le O(\sigma^2 w) << d^{-\varepsilon}\|C_1(t)\|_{op}$.

It is true that the error does not vanish. However, for our purpose it suffices to show that $\|\hat{C}_1 - C_1\|_{op}$ is infinitely smaller than $C_1$ for all times, regardless of the magnitude of $\|C_1(t)\|_{op}$.

We see how both the lazy and balanced dynamics appear as special cases depending on how large the variance at initialization $\sigma^2$ is in comparison to the singular values of the matrix $A_{\theta(t)}$:

- **Lazy:** When $\sigma^2 w \gg s_{max}(A_{\theta(t)})$, then $\hat{C}_1 \approx \sigma^2 w I_{d_{out}}$ and $\hat{C}_2 \approx \sigma^2 w I_{d_{in}}$, recovering the lazy dynamics.

- **Balanced:** When $\sigma^2 w \ll s_{min}(A_{\theta(t)})$, then $\hat{C}_1 \approx \sqrt{A_{\theta(t)}^T A_{\theta(t)}}$ and $\hat{C}_2 \approx \sqrt{A_{\theta(t)} A_{\theta(t)}^T}$, recovering the balanced dynamics.

But clearly there can be times when neither conditions are satisfied, when some singular values of $A_{\theta(t)}$ are larger than the threshold $\sigma^2 w$ while others are smaller, in such cases we are in a mixed regime, where the network is lazy along the small singular values of $A_{\theta(t)}$ ($s_i \ll \sigma^2 w$) and active/balanced along the large ones ($s_i \gg \sigma^2 w$).

At initialization, the singular values are of size $\sigma^2\sqrt{wd}$. This implies that with overparametrization ($w \gg d$), all singular values start in the lazy regime and follow the simple lazy dynamics, which may (or may not) lead to some singular growing and crossing the $\sigma^2 w$ threshold, at which point they will switch to balanced dynamics (after a short transition period when the singular value is around the threshold $s_i \approx \sigma^2 w$). Once a singular value is far past the threshold $s_i \gg \sigma^2 w$, training along this singular value will be much faster than along the lazy singular values (this speed up can be seen in Figure 1). This allows the newly active singular values to converge while the lazy singular values remain almost constant. Once the active singular values have converged, the slow training of the remaining lazy singular values continues until some of these singular values reaches the threshold, or until GD converges.

This type of behavior is illustrated by the following formula, which describes the derivative in time of the $i$-th singular value $s_{i,t}$ of $A_t$, with singular vectors $u_{i,t}, v_{i,t}$:

$$s_{i,t+1} - s_{i,t} \approx \eta_t u_{i,t}^T \partial_t A_{\theta(t)} v_{i,t} \approx -2\eta_t \sqrt{s_{i,t}^2 + \sigma^4 w^2}\, u_{i,t}^T \nabla C(A_{\theta(t)}) v_{i,t},$$

where the prefactor $2\eta_t \sqrt{s_{i,t}^2 + \sigma^4 w^2}$ describes the effective learning rate along the $i$-th singular value, which depends on the $i$-th singular value $s_{i,t}$ itself.

This suggests that it is more natural to distinguish between the lazy and active regime at a much more granular level: at every time $t$ a singular value can be either active or lazy (or very close to the transition but this typically only happens for a very short time). In contrast, the traditional definition of the lazy regime was defined for a whole network and over the whole training time. To avoid confusion, we call this the pure lazy regime, where all singular values remain lazy throughout training. This begs the question of whether a pure balanced regime also exists, but all singular values will always be lazy for at least a short time period (assuming $w > d$), and as we will see this short lazy period plays a crucial role in aligning the network so that the subsequent balanced regime can learn successfully. A pure balanced regime can only be obtained in the underparametrized regime, or by taking a balanced initialization instead of the traditional i.i.d. random initialization.

While this challenges the traditional lazy/active dichotomy, it also reinforces it, as it shows that there is no fundamentally different third regime, only lazy, active, and some mix of the two. Theorem 1 thus allows us to revisit previous descriptions of lazy and balanced dynamics and 'glue them together' to extend them to the general case. This simple strategy will allow to almost fully 'fill in the phase diagram', i.e. describe the dynamics, convergence and generalization properties of DLNs for almost all reasonable initialization scales $\sigma^2$ and widths $w$.

*Remark.* The transition of a singular value $s_i$ from lazy to active can be understood as a form of alignment happening in the hidden layer: the two vectors $W_1 v_i$ and $W_2^T u_i$ for $u_i, v_i$ the left and right singular vectors of $s_i$ are orthogonal in the lazy regime and become perpendicular in the balanced

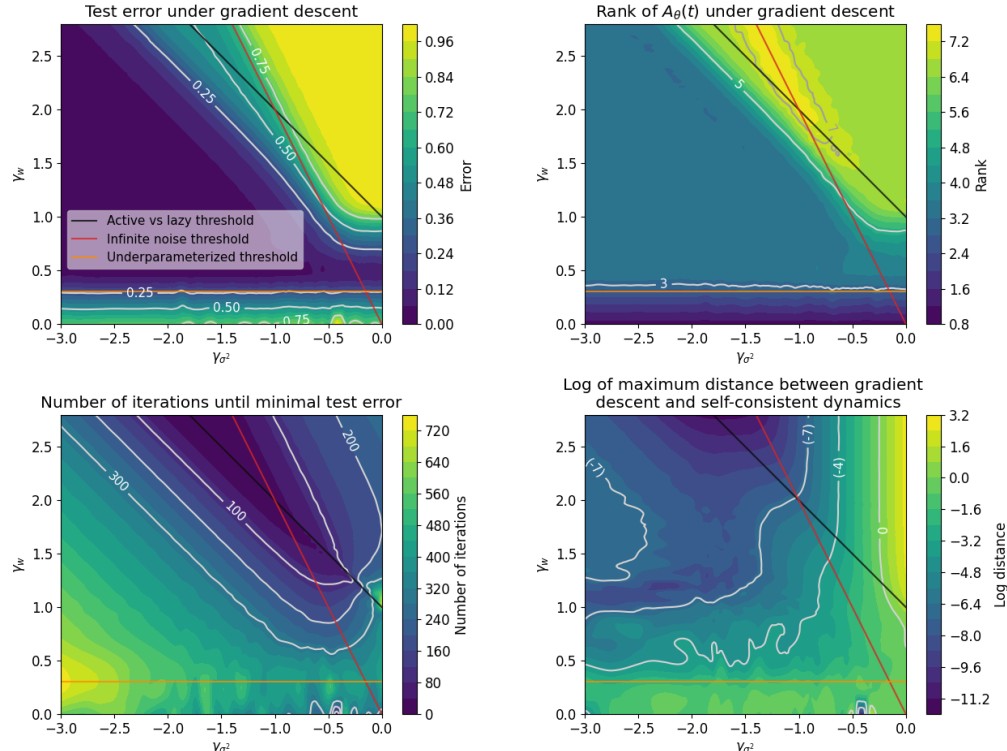

Figure 2: As a function of $\gamma_{\sigma^2}, \gamma_w$, we run GD and plot different quantities. Our theoretical results only apply to the top left region for $\gamma_w > 1$ and below the red line, although these plots suggest that some results may extend to smaller $\gamma_w$s. (Top left panel): We plot the smallest test error $\frac{1}{d^2}\|A_{\theta(t)} - A^*\|_F^2$ in the whole run. The active region (below the black line) has a small error while the lazy region does not. (Top right panel): We plot the stable rank of $A_{\theta(t)}$ (defined as $\|A_{\theta(t)}\|_F^2/\|A_{\theta(t)}\|_{op}^2$) at the time of minimal test error. In this experiment, we took $\text{Rank}A^* = 5$. We see that the active region has approximately the correct rank while the lazy region overestimates it. (Bottom left panel): We plot the number of iterations until minimal test error, illustrating the trade-off between test error and training time. (Bottom right panel): We compute $\ln\left(\frac{1}{d^2}\|A_{\theta(t)} - \hat{A}_{\theta(t)}\|_F^2\right)$ where $A_{\theta(t)}$ comes from GD and $\hat{A}_{\theta(t)}$ from the self-consistent dynamics. We observe that this distance is not only small for the region where our theoretical results apply but also almost everywhere outside this region.

regime. Indeed the normalized scalar product of these two vectors satisfies

$$\frac{u_i^T W_2 W_1 v_i}{\|W_2^T u_i\| \|W_1 v_i\|} = \frac{s_i}{\sqrt{u_i^T C_2 u_i}\sqrt{u_i^T C_1 u_i}} \approx \frac{s_i}{\sqrt{s_i^2 + \sigma^4 w^2}}$$

which is close to zero for lazy singular values $s_i \ll \sigma^2 w$ and close to one for active ones $s_i \gg \sigma^2 w$.

## 2.1 Phase Diagram for MSE

To illustrate the power of Theorem 1 we provide a phase diagram of the behavior of large shallow networks on a MSE task, for almost all (reasonable) choices of width $w$ and variance $\sigma^2$ scalings.

We want to recover a rank $K$ and $d \times d$-dimensional matrix $A^*$ with $s_i(A^*) = da_i$ for some $a_1 \geq a_2 \geq \cdots \geq a_K$ independent of the dimension $d$. We however only observe a noisy version $A^* + E$ for some $E$ such that $\|E\|_{op} \leq c_0 d^\delta$. One could imagine $E$ to have iid random Gaussian entries $\mathcal{N}(0, 1)$ in which case $\|E\|_{op} \leq c_0\sqrt{d}$ with high probability.

As the dimension $d$ grows, the size of the network needs to scale too, as well as the initialization variance, but it is unclear what is the optimal way to choose $w$ and $\sigma^2$. We will therefore consider general scalings $w = d^{\gamma_w}$ and $\sigma^2 = d^{\gamma_{\sigma^2}}$. We will now describe the $(\gamma_w, \gamma_{\sigma^2})$-phase diagram which features 4 regimes: underparametrized, infinite-noise, lazy and mixed/active.

We can identify a region of 'reasonable' pairs $(\gamma_{\sigma^2}, \gamma_w)$ by ruling out degenerate behavior. First, the width $w$ needs to be larger than the dimension $d$, since a network of width $w$ can only represent matrices of rank $w$ or less, this means that we need $\gamma_w \geq 1$. Another constraint comes from the variance of $A_\theta$ at initialization: the entries $A_{\theta(0),ij}$ at initialization have variance $\sigma^4 w$. We want this variance to go to zero as $d$ grows which implies that we need $2\gamma_{\sigma^2} + \gamma_w < 0$.

Now within this reasonable region we observe two regimes, the pure lazy regime for $1 < \gamma_{\sigma^2} + \gamma_w$ where the network simply fits $A^* + E$ thus failing to learn $A^*$ and the mixed regime for $1 > \gamma_{\sigma^2} + \gamma_w$ where the dynamics are lazy for a short amount of time until $K$ singular values grow large enough to switch to the balanced dynamics and fit the true matrix $A^*$.

**Theorem 2.** *For pairs $\gamma_w, \gamma_{\sigma^2}$ such that $\gamma_w > 1$ and $2\gamma_{\sigma^2} + \gamma_w < 0$, we have two regimes:*

- **Lazy** *($1 < \gamma_{\sigma^2} + \gamma_w$): with a learning rate $\eta \ll \frac{d^2}{\sigma^2 w}$ we have that for all time $t$,*
  $\frac{1}{d^2} \left\| A_{\theta(t)} - A^* \right\|_F^2 \geq c.$

- **Active** *($1 > \gamma_{\sigma^2} + \gamma_w$): with a learning rate $\eta \ll \frac{d^2}{s_1(A^*)} \sim d$, and at time*

$$t = \frac{1}{\eta}\left( \frac{\Delta}{a_K} + \frac{2max(1, 2\Delta)}{c(a_1, \ldots, a_K)} + \frac{max(1, 2\Delta)}{2a_K} \right) d\log d + \eta^{-1}O(d\log\log d),$$

*for $\Delta = 1 - \gamma_{\sigma^2} - \gamma_w > 0$, we have that*

$$\frac{1}{d^2}\left\| A_{\theta(t)} - A^* \right\|_F^2 \leq O(\sigma^4 w + \frac{\sigma^4 w^2 \log^2 d}{d^2} + d^{-\frac{1}{2}} + \frac{\sigma^2 w}{d} + \eta^2 \frac{\log^2 d}{d^2}),$$

*for $c(a_1, \ldots, a_K) = \frac{min_{k,j:a_k \neq a_j}|a_k - a_j|a_K^2}{max_{k,j:a_k \neq a_j}\left|a_k^2 - a_j^2\right|}.$*

Note that all the terms inside the final $O(\ldots)$ term vanish: $\sigma^4 w \to 0$ because $\gamma_{\sigma^2} + \gamma_w < 0$, $\frac{\sigma^4 w^2 \log^2 d}{d^2} + \frac{\sigma^2 w}{d} \to 0$ since $1 > \gamma_{\sigma^2} + \gamma_w$, and $\eta^2 \frac{\log^2 d}{d^2} \to 0$ since we assumed $\eta \ll d$.

This shows that the lazy regime only appears for very large widths $\gamma_w > 2$ (or at least the lazy regime with finite variance at initialization). Indeed the choice $\gamma_w = 2, \gamma_{\sigma^2} = -1$ is at the boundary of the lazy regime with the smallest $\gamma_w$. This could explain why it is rare to observe the lazy regime in practice.

Our theoretical results applies to the overparametrized regime $w \gg d$, but actually we only want to fit $A^*$ which has a much smaller rank $r$, and so we might only need $w \gg r$. Figure 2, top left panel, confirms this, since we see a good generalization even for small widths $w < d$, and in particular when $w \approx \text{Rank}A^*$. But to leverage this underparametrized regime, one would need to know the rank of the true matrix $A^*$ in advance, which is typically not the case in practice. Nevertheless, the interesting behavior we observe in the (mildly) underparametrized regime warrants further analysis, and the fact that our self-consistent dynamics remain a good approximation in this regime (Figure 2, bottom right panel), suggests that the analysis we present here could be extended to this regime too.

Finally, we observe a trade-off between generalization error and training time: on one hand the test error has terms that scale negatively with $1 - \gamma_{\sigma^2} - \gamma_w$, which is the distance to the lazy/active transition, on the other hand, the time it takes to reach the minimal loss point scales positively with the same term. This can be seen from Figure 2, bottom left panel, which plots the number of steps required to reach minimal test error, which increases as one goes further into the active regime.

*Remark.* In general when trying to fit a matrix $B$ (instead of the special case $B = A^* + E$), the transition between lazy and mixed regime is when $\sigma^2 w \approx \|B\|_{op}$. Thus the exact location of the transition is task-dependent, so that the same variance $\sigma^2$ and width $w$ can lead to NTK or mixed regimes depending on the task. For example, let us assume that $A^*$ is full-rank instead of finite rank, then we expect $\|A^*\|_{op} \sim \sqrt{d}$ instead of $\|A^*\|_{op} \sim d$, thus the transition would be at $\frac{1}{2} = \gamma_{\sigma^2} + \gamma_w$ instead of $1 = \gamma_{\sigma^2} + \gamma_w$. This suggests that linear networks are able to adapt themselves to the task:

leveraging active dynamics when the true data is low-rank to get better generalization, or remaining in the lazy dynamics in the absence of low-rank structure, to take advantage of the faster convergence. Note also that in the absence of sparsity, the lazy regime can be attained with a smaller width ($\gamma_w > 1$ instead of $\gamma_w > 2$), since the choice $\gamma_w = 1, \gamma_{\sigma^2} = -\frac{1}{2}$ is already on the boundary of the lazy regime.

## 3   Conclusion

We prove a surprisingly simple self-consistent dynamic for the evolution of the matrix represented by a shallow linear network under gradient descent. This description not only unifies the already known lazy and balanced dynamics, but reveals the existence of a spectrum of mixed dynamics where some of the singular values are lazy while others are balanced.

Thanks to this description we are able to give an almost complete phase diagram of training dynamics as a function of the scaling of the width and variance at initialization w.r.t. the dimension.

A natural question that comes out of these results is whether nonlinear network also feature similar mixed regimes, and whether they could be the key to understand the convergence of general DNNs.

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

The appendix is structured as follows.

- In section A, we introduce the notation, and establish several results about how perturbing a matrix would impact its singular vectors.

- In section C, we study the gradient flow dynamics of $A_t$ in the active regime and prove that $A_t$ will be approximately aligned with $A^*$ throughout the Saddle-to-Saddle regime.

- In section B, we prove theorem 1 for general cost.

- In section D, we study the gradient flow dynamics for $A_{\theta(t)}$ in the lazy regime.

- In section E, we show that $A_{\theta(t)}$ is also approximately aligned with $A^*$ throughout the Saddle-to-Saddle regime, using results in section C and section B. In subsection E.3, we summarize the approximate dynamics of $A_{\theta(t)}$ throughout training. In section E.4, we bound the final test error.

- In section F, we bound the error from gradient descent and prove theorem 2.

- In section G, we describe the experimental setup.

## A  Preliminaries

### A.1  Convention and Notation

**Constants**. $d$ is the dimension of the input and the output layer, $K$ is the rank of matrix $A^*$, and $w$ is the dimension of hidden layer. $c$ and $C$ will usually denote constants that are independent of $d$, and depending on the context, the value of $c$ and $C$ might be different. If $x$ is a scalar that depends on $d$ and $y$ is a scalar, then $x = O(y)$ means there exists a constant $c$ independent of $d$, such that for $d$ sufficiently large we have $x \leq cy$. If $A$ is a matrix, then $A = O(y)$ means there exists a constant $c$ independent of $d$ such that for $d$ sufficiently large, $\|A\|_{op} \leq cy$. $O(y)$ can be either a matrix or a scalar, and its meaning will be always clear from context.

**Matrix**. We use $\| \cdot \|_{op}, \| \cdot \|_F, \| \cdot \|$ to denote the operator norm of a matrix, the Frobenius norm of a matrix, and the $L^2$ norm for vectors. For every matrix $A$, we define max$A$ as the $L^\infty$ norm, $\max_{i,j} |A_{ij}|$. We use $I$ to denote identity matrix, the dimension of which is determined by context. We shall assume that the signal singular values of $A^*$ are $s_1^*, \ldots, s_K^*$. $\forall i = 1, 2, \ldots, K$, and $s_i^* = a_i d$ where $a_1 \geq a_2 \geq \ldots \geq a_K$ are constants independent of $d$. By selecting proper basis in the input and output space, we assume that $A^* = S^*$, where $S^*$ is the diagonal matrix consisting of singular values of $A^*$, ordered from largest to smallest. The $p, q$-th element of a matrix $A$ is denoted $A_{pq}$. We reserve the notation $A(i, j)$ for the $i, j$-th block matrix of $A$, which we shall define below.

**Submatrix**. Assume that $n_0 = 0$, and $a_1 = \ldots = a_{n_1} > a_{n_1+1} = \ldots a_{n_2} > \ldots = a_{n_m} = a_K$, and let $n_{m+1} = d$. For a matrix $U$, we define the $k, j$-th sub-block $U(k, j)$ of a matrix $U$ as $U_{n_{k-1}+1:n_{k+1}, n_{j-1}+1:n_j}$, with both sides included. Notice that $U^T(k, j) = U(j, k)^T$. In this notation, we can write the singular value decomposition of a matrix $A$ as

$$A(i, j) = \sum_{k:\text{signal}} U(i, k)S(k, k)V(j, k) + U(i, m + 1)S(m + 1, m + 1)V^T(m + 1, j).$$

We call an index $k$ (of sub-block) "signal", if $k \leq m$. Index $m + 1$ is called "noise". Let $S(k, k)^*$ be block matrix $A^*(n_k : n_{k+1}, n_k : n_{k+1})$. Then $A^* = \text{diag}(S(1, 1)^*, \ldots, S(m, m)^*)$ and $S(k, k)^* = s_{n_k} I$ is the $k$-th sub-block of $A^*$. Each matrix has only finitely many sub-blocks.

**Indexing Conventions**. Entries of matrices will usually be indexed by $p, q, r$ and sub-blocks of matrices will usually be indexed by $i, j, k, \ell$. Usually $k$ ranges from 1 to $m$, and $j$ usually ranges from 1 to $m + 1$.

**Element-wise Product**. We use $\odot$ to represent element-wise product of two matrix of the same shape.

**Important Assumptions**. Throughout the paper, we shall always assume that

1. $\gamma_w > 1$ (i.e., $w >> d$).
2. $2\gamma_{\sigma^2} + \gamma_w < 0$. (i.e., $\sigma^4 w << 1$).

## A.2 Matrix Inequalities

In the proof of main theorems, we will work extensively with inequalities of matrix norms and inequalities that involves element-wise product. The element-wise product appears naturally in the derivative of singular vectors of a matrix.

**Lemma A.1.** *Assume that A, B and R are square matrices. Let $R_{max} = max_{i,j}|R_{ij}|$. Then*

$$tr[A(R \odot B)] = tr[BR^T \odot A],$$

*and*

$$|tr[A(R \odot B)]| \leq R_{max}\sqrt{tr(A^T A)}\sqrt{tr(B^T B)}$$

*In particular, if $\forall p, q, R_{pq} \geq R_{min} > 0$, then*

$$tr[A^T(R \odot A)] \geq R_{min}tr(A^T A)$$

*Proof.* All are simple computations.

$$tr[AR \odot B] = \sum_{p,q} A_{pq}R_{qp}B_{qp}$$

$$tr[BR^T \odot A] = \sum_{p,q} B_{pq}R_{pq}A_{qp}$$

The two equations above prove the first claim.

$$|tr[A(R \odot B)]| \leq \sqrt{tr(A^T A)}\sqrt{tr((R \odot B)^T R \odot B)}$$
$$\leq R_{max}\sqrt{tr(A^T A)}\sqrt{tr(B^T B)}$$

This completes the second claim.

$$tr[A^T(R \odot A)] = \sum_{i,j} A_{ji}R_{ji}A_{ji}$$
$$\geq R_{min}tr[A^T A]$$

This completes the third claim. □

**Lemma A.2.** $\sigma_{min}(A)\|B\|_F \leq \|AB\|_F \leq \sigma_{max}(A)\|B\|_F$.

*Proof.* This is lemma B.3 of [54]. □

## A.3 Perturbation of Singular Values and Singular Vectors

We will often use the following variant of the Davis-Kahan $\sin\theta$ theorem.

**Theorem A.3** (DK-$\sin\theta$ Theorem). *. Let $\Sigma, \hat{\Sigma} \in \mathbb{R}^{p \times p}$ be symmetric, with eigenvalues $\lambda_1 \geq \ldots \geq \lambda_p$ and $\hat{\lambda}_1 \geq \ldots \geq \hat{\lambda}_p$. Fix $1 \leq r \leq s \leq p$, let $d = r - s + 1$ and let $V = (v_r, \ldots, v_s)$ and $\hat{V} = (\hat{v}_r, \ldots, \hat{v}_s)$ have orthonormal columns satisfying $\Sigma v_j = \lambda_j v_j$ and $\Sigma \hat{v}_j = \hat{\lambda}_j v_j$. Let $\sigma_1, \ldots, \sigma_d$ be the singular values of $\hat{V}^T V$. Let $\Theta(V, \hat{V})$ be the diagonal matrix with $\cos\Theta(V, \hat{V})_{jj} = \sigma_j$ and $\sin\Theta(V, \hat{V})$ be defined entry-wise. Then*

$$\|\sin\Theta(V, \hat{V})\|_F \leq \frac{2min(\sqrt{d}\|\hat{\Sigma} - \Sigma\|_{op}, \|\hat{\Sigma} - \Sigma\|_F)}{min(\lambda_{r-1} - \lambda_r, \lambda_s - \lambda_{s+1})}.$$

*Proof.* This is theorem 2 in [53]. □

The implication of the theorem is that if two matrices are sufficiently close, then their singular vectors are also close to each other. In the case where $r = s$ and $\lambda_r$ is of multiplicity 1, the theorem reduces to saying that the sine value of the angle between $v_r$ and $\hat{v}_r$ is very small.

The term $\|\sin\Theta(V, \hat{V})\|_F$ is complicated to take derivative. In this paper we will use the following characterization of alignment, which is easier to take derivatives.

**Lemma A.4.** *Let $\hat{\Sigma}$ be a $d \times d$ diagonal matrix, and let $s_1, \ldots, s_K, \ldots, s_d$ be its diagonal entries. Assume that $s_1 = \ldots = s_{n_1} > s_{n_1+1} = \ldots = s_{n_2} \geq \ldots s_{n_m} = s_K > s_{K+1} = \ldots = s_d$. Then $\hat{\Sigma}$ has $m + 1$ blocks in total. Assume that $\|X - \hat{\Sigma}\|_{op} = d^\alpha$. Let $X = USV^T$, and define*

$$x = 4K - \sum_{k:signal} \text{tr}(U^T(k, k)U(k, k) + V^T(k, k)V(k, k) + 2U(k, k)V(k, k)^T).$$

*Then*

$$x \leq K\frac{\|\hat{\Sigma} - X\|_{op}}{s_K} + \frac{ms_1}{s_K}\left(\frac{2min(\sqrt{K}(\|X\|_{op} + \|\hat{\Sigma}\|_{op}), \|X\|_F + \|\hat{\Sigma}\|_F)}{min_k(s_{n_k}^2 - s_{n_{k+1}}^2)}\right)^2 \|X - \hat{\Sigma}\|_{op}^2$$

**Remark.** *To better understand $x$, consider the special case where each signal singular value is of multiplicity 1. Since matrix $X$ is "close" to diagonal matrix $\hat{\Sigma}$, matrix $U$ and matrix $V$ should also be close to identity along signal directions: $|U_{kk}| \approx 1$, $|V_{kk}| \approx 1$, $U_{kk}V_{kk} \approx 1$ for $1 \leq k \leq K$. Quantity $x$ captures how much $|U_{kk}|, |V_{kk}|, U_{kk}V_{kk}$ deviate from 1.*

*Proof.* Let $X = USV^T$. Then $X^TX = VS^2V^T$ is a symmetric matrix. Let $V(:, k) = (v_{n_{k-1}+1}, \ldots, v_{n_k})$ be the singular vectors of $X$ corresponding to $s_{n_{k-1}+1}, \ldots, s_{n_k}$. There is some freedom to choose $\hat{V}$, but for simplicity we pick $\hat{V}(:, k) = (e_{n_{k-1}+1}, \ldots, e_{n_k})$ where $e_i$ is the $i$-th coordinate vector. As a result, $\hat{V}(:, k)^TV(:, k) = V(k, k)$. Apply DK-sin $\theta$ theorem to $V$ and $\hat{V}$, we see that

$$\|\sin\Theta(V(:, k), \hat{V}(:, k))\|_F \leq \frac{2min(\sqrt{n_k - n_{k-1}}\|X^TX - \hat{\Sigma}^2\|_{op}, \|X^TX - \hat{\Sigma}^2\|_F)}{min(s_{n_{k-1}}^2 - s_{n_{k-1}+1}^2, s_{n_k}^2 - s_{n_k+1}^2)}$$

Let $\sigma_p$ be the singular values of $V(k, k)$, for $1 \leq p \leq n_k - n_{k-1}$. Here we are using the notation for sub-block of a big matrix. Then $\sin\Theta(V(:, k), \hat{V}(:, k))$ is the diagonal matrix, whose diagonal entries are given by $\sqrt{1 - \sigma_p^2}$. So $\|\sin\Theta(V, \hat{V})\|_F^2 = \sum_p(1 - \sigma_p^2) = tr[I - V(k, k)^TV(k, k)]$. Now observe that

$$\|X^TX - \hat{\Sigma}^2\|_{op} \leq (\|X\|_{op} + \|\hat{\Sigma}\|_{op})\|X - \hat{\Sigma}\|_{op}$$

$$\|X^TX - \hat{\Sigma}^2\|_F \leq (\|X\|_F + \|\hat{\Sigma}\|_F)\|X - \hat{\Sigma}\|_{op}$$

We conclude that

$$tr[I - V(k, k)^TV(k, k)] \leq \left(\frac{2min(\sqrt{n_k - n_{k-1}}(\|X\|_{op} + \|\hat{\Sigma}\|_{op}), \|X\|_F + \|\hat{\Sigma}\|_F)}{min(s_{n_{k-1}}^2 - s_{n_{k-1}+1}^2, s_{n_k}^2 - s_{n_k+1}^2)}\|X - \hat{\Sigma}\|_{op}\right)^2.$$

Similar conclusion is true for $U(k, k)$. Next we bound $tr(I - U(k, k)V(k, k)^T)$.

$$\|\hat{\Sigma}(k, k) - X(k, k)\|_F^2 = \sum_{q_1,q_2=n_{k-1}+1}^{n_k}(\hat{\Sigma}_{q_1q_2} - X_{q_1q_2})^2 \leq (n_k - n_{k-1})\|\hat{\Sigma} - X\|_{op}^2.$$

$$\|\hat{\Sigma}(k,k) - X(k,k)\|_F^2 = \|\hat{\Sigma}(k,k) - \sum_j U(k,j)S(j,j)V^T(j,k)\|_F^2$$

$$= \|\hat{\Sigma}(k,k) - \sum_{j\neq k} U(k,j)S(j,j)V(k,j)^T - U(k,k)S(k,k)V(k,k)^T\|_F^2$$

$$\geq \|\hat{\Sigma}(k,k) - U(k,k)\hat{\Sigma}(k,k)V(k,k)^T\|_F^2$$
$$- \|\sum_{j\neq k} U(k,j)S(j,j)V(k,j)^T\|_F^2$$
$$- \|U(k,k)(S(k,k) - \hat{\Sigma}(k,k))V(k,k)^T\|_F^2$$

$$\geq s_{n_k}^2 \|I - U(k,k)V(k,k)^T\|_F^2$$
$$- \sum_{j\neq k} \|U(k,j)\|_F^2 \|S(j,j)V(k,j)\|_{op}^2$$
$$- \|X - \hat{\Sigma}\|_{op}^2$$

$$\geq s_{n_k}^2 \|I - U(k,k)V(k,k)^T\|_F^2$$
$$- s_1^2 tr(I - U(k,k)^T U(k,k)) tr(I - V(k,k)^T V(k,k))$$
$$- \|X - \hat{\Sigma}\|_{op}^2$$

For every $n_k - n_{k-1} \times n_k - n_{k-1}$ matrix $M$, we have $tr(A) \leq \sum_p |\lambda_p| \leq \sqrt{n_k - n_{k-1}} \left(\sum |\lambda_p|^2\right)^{\frac{1}{2}} = \sqrt{n_k - n_{k-1}}\|M\|_F$. Using this inequality, we conclude that

$$tr(I - U(k,k)V(k,k)^T) \leq (n_k - n_{k-1})\frac{\|\hat{\Sigma} - X\|_{op}}{s_{n_k}}$$
$$+ \frac{s_1}{s_{n_k}} \left(\frac{2\min(\sqrt{n_k - n_{k-1}}(\|X\|_{op} + \|\hat{\Sigma}\|_{op}), \|X\|_F + \|\hat{\Sigma}\|_F)}{\min(s_{n_{k-1}}^2 - s_{n_{k-1}+1}^2, s_{n_k}^2 - s_{n_k+1}^2)}\|X - \hat{\Sigma}\|_{op}\right)^2$$

Summing on $k$, we conclude that

$$x \leq K\frac{\|\hat{\Sigma} - X\|_{op}}{s_K} + \frac{ms_1}{s_K} \left(\frac{2\min(\sqrt{K}(\|X\|_{op} + \|\hat{\Sigma}\|_{op}), \|X\|_F + \|\hat{\Sigma}\|_F)}{\min_k(s_{n_k}^2 - s_{n_{k+1}}^2)}\right)^2 \|X - \hat{\Sigma}\|_{op}^2$$

$\square$

In the case where $d \to \infty$, $m$ is a constant, $\alpha < 1$, $s_1 = O(s_K)$, $s_K > cs_{K+1}$, $K$ is a constant, $\|X - \hat{\Sigma}\|_{op} = o(\|\hat{\Sigma}\|_{op})$, and $\min_k(s_{n_k}^2 - s_{n_{k+1}}^2) \geq cs_K^2$ for some constant $c$, we have $x \leq O(\frac{\|X-\hat{\Sigma}\|_{op}}{\|\hat{\Sigma}\|_{op}})$.

**Lemma A.5.** *Let $X = USV^T$, and define*

$$x = 4K - \sum_{k:signal} tr(U^T(k,k)U(k,k) + V^T(k,k)V(k,k) + 2U(k,k)V(k,k)^T).$$

*Let $\Sigma$ be a $d \times d$ diagonal matrix whose diagonal entries are given by $b_1, \ldots, b_K, 0, \ldots, 0$, where $b_1 = \ldots b_{n_1} > b_{n_1+1} = \ldots = b_{n_2} \geq \ldots = b_{n_m} = b_K$. Then*

$$\|\Sigma - U^T\Sigma V\|_{op} \leq \|\Sigma\|_{op}((m+1)^3\sqrt{x} + 2(m+1)^3 x)$$

*Proof.* First observe that $\|\Sigma - U^T\Sigma V\|_{op} \leq \sum_{i,j=1}^{m+1} \|\Sigma(i,j) - U^T\Sigma V(i,j)\|_{op}$. If $i \neq j$, then $\Sigma(i,j) = 0$,

$$\|U^T\Sigma V(i,j)\|_{op} = \| \sum_{\ell=1}^{m+1} U(\ell,i)^T\Sigma(\ell,\ell)V(\ell,j)\|_{op}$$

$$\leq b_1 \sum_{\ell=1}^{m+1} \|U(\ell,i)\|_{op}\|V(\ell,j)\|_{op}$$

$$\leq b_1(m+1)\sqrt{x}.$$

If $i = j$, then $\Sigma(i,i) = b_{n_i}I$,

$$\|U^T\Sigma V(i,i) - b_{n_i}I\|_{op} \leq \|b_{n_i}U(i,i)^TV(i,i) - b_{n_i}I\|_{op} + b_1\sum_{\ell \neq i} \|U(\ell,i)\|_{op}\|V(\ell,i)\|_{op}$$

$$\leq b_1(m+1)x + b_1(m+1)x.$$

Therefore

$$\|\Sigma - U^T\Sigma V\|_{op} \leq \|\Sigma\|_{op}((m+1)^3\sqrt{x} + 2(m+1)^3 x).$$

$\square$

This bound is not optimal in $m$, but throughout the paper, $m$ is of constant order and it is fine to miss a constant factor when estimating error.

# B    Proof of Theorem 1

## B.1    Weak bound

We prove the following weak bound.

**Proposition B.1.** *For every $\varepsilon > 0$ and every $t < T$, we have with high probability,*

$$\|W_1^T W_1 - \sqrt{A^T A + \sigma^4 w^2 I}\|_{op} \leq (1+\varepsilon)\sigma^2 w \qquad (3)$$

*Analogous results holds for $W_2$.*

Our main tool is the following lemma.

**Lemma B.2.** *For every cost $C$ and every time $t$, we have*

$$W_2^T W_2(t) - W_1 W_1^T(t) = W_2^T(0)W_2(0) - W_1(0)W_1^T(0) \qquad (4)$$

*Proof.* Let $L = \|A^* - A\|^2$ be the loss function. Then

$$\partial_t W_1 = 2W_2^T\nabla C$$
$$\partial_t W_2 = 2\nabla C W_1^T$$

We see that

$$\partial_t(W_1 W_1^T) = 2W_2^T\nabla C W_1^T + W_1\nabla C^T W_2 = \partial_t(W_2^T W_2)$$

$\square$

Next, we show that at initialization, $W_1 W_1^T$ and $W_2^T W_2$ are approximately orthogonal projections, up to a factor. Stating precisely, we have the following lemma.

**Lemma B.3.** *There exists two projections $P_1$ and $P_2 : \mathbb{R}^w \to \mathbb{R}^w$ such that the following are true.*

1. *The image of $P_1$ and $P_2$ are orthogonal to each other;*

2. *With high probability,*

$$\|W_1 W_1^T(0) - \sigma^2 w P_1\|_{op} = O(\sigma^2\sqrt{wd}\log d) \qquad (5)$$

$$\|W_2^T W_2(0) - \sigma^2 w P_2\|_{op} = O(\sigma^2\sqrt{wd}\log d) \qquad (6)$$

*Proof of Lemma B.3* . : At initialization, the rank of $w$is $w$ with probability 1. Therefore $W_1 W_1^T$ has $w - d$ eigenvalues that are 0, and the $w$ non-zero eigenvalues equals the eigenvalues of $d \times d$ matrix $W_1^T W_1$.

$W_1^T W_1(0)$ is a (scaled) Wishart ensemble, whose limiting distribution is given by the Marchenko-Pastur law. The Marchenko-Pastur law, as stated in [4], proves the following. Let $X$ be an $M \times N$ matrix with complex-valued independent entries $X_{i\mu}$ such that

1. $\mathbb{E}[X_{i\mu}] = 0$;

2. $\mathbb{E}[|X_{i\mu}|^2] = \frac{1}{\sqrt{MN}}$;

3. for every $p \in \mathbb{N}$, there exists a constant $C_p$ such that

$$\mathbb{E}\left[\left|(NM)^{\frac{1}{4}} X_{i\mu}\right|^p\right] \leq C_p.$$

Here, $M$ satisfies

$$0 < C^{-1} \leq \frac{\log M}{\log N} \leq C < \infty$$

for some constant $C$ independent of $M$ and $N$. Let $\phi = \frac{M}{N}$, which may or may not depend on $N$. Then the eigenvalues of $N \times N$ matrix $X^* X$ has the same asymptotics as

$$\rho_\phi(dx) := \frac{\sqrt{\phi}}{2\pi} \sqrt{\frac{[(x - \gamma_-)(\gamma_+ - x)]_+}{x^2}} dx + (1 - \phi)_+ \delta(dx)$$

where

$$\gamma_\pm := \sqrt{\phi} + \frac{1}{\sqrt{\phi}} \pm 2 \tag{7}$$

In our situation, $W_1$ is a $w \times d$ matrix with independent and identically distributed Gaussian entries whose variance is $\sigma^2$. Let $M = w$, $N = d$ and therefore $W_1^T W_1(0)$ has the same distribution as $\sigma^2 \sqrt{wd} X^* X$. Notice that for this choice of $M$ and $N$, the asymptotic distribution eigenvalues of $X^* X$ is $\rho_{\frac{w}{d}}(dx)$. Notice that $\rho_{\frac{w}{d}}$ is supported on interval $\left[\sqrt{\frac{w}{d}} + \sqrt{\frac{d}{w}} - 2, \sqrt{\frac{w}{d}} + \sqrt{\frac{d}{w}} + 2\right]$, from which we conclude that in the limit, all eigenvalues of $W_1^T W_1(0)$ is approximately $\sqrt{\frac{d}{w}}$.

By Theorem 2.10 of [4], we have eigenvalue rigidity results for $X^* X$. Let $\lambda_k'$ be the $k$-th largest eigenvalue for $X^* X$. $\forall k \in \{1, 2, \ldots, w\}$, we have

$$|\lambda_k' - \gamma_k| < d^{-\frac{2}{3} + \varepsilon} \tag{8}$$

with high probability. Here $\gamma_\alpha$ is defined through

$$\int_{\gamma_k}^\infty \rho_\phi(dx) = \frac{k}{d} \tag{9}$$

Let $\lambda_k$ be the $k$-th largest eigenvalue of $W_1^T W_1(0)$. By the relationship between $W_1^T W_1(0)$ and $X^* X$, we know $\lambda_k$ has the same law as $\sigma^2 \sqrt{wd} \lambda_k'$. Then for every $k$,

$$\left|\lambda_k - \sigma^2 w\right| \leq \sigma^2 \sqrt{wd} \left(|\lambda_k' - \gamma_k| + \left|\gamma_k - \sqrt{\frac{w}{d}}\right|\right) \tag{10}$$

$$\leq O(\sigma^2 \sqrt{wd}) \tag{11}$$

with high probability. We conclude that the first $w$ eigenvalues of $W_1^T W_1(0)$ is at most $O(\sigma^2 \sqrt{wd})$-away from 1 and all other eigenvalues are 0. Therefore there exists a projection $P_1$ such that

$$\|W_1 W_1^T(0) - \sigma^2 w P_1\|_{\text{op}} = O(\sigma^2 \sqrt{wd}) \tag{12}$$

Similarly, there exists a projection $\tilde{P}_2$ such that

$$\|W_2^T W_2(0) - \sigma^2 w \tilde{P}_2\|_{\text{op}} = O(\sigma^2 \sqrt{wd}) \tag{13}$$

Notice that $\tilde{P}_2$ is not exactly orthogonal to $P_1$, and it remains to find a projection $P_2$ that is orthogonal to $P_1$ and is close to $W_2^T W_2(0)$. Assume that the column vectors of $W_1(0)$ are $u_1, \ldots, u_d \in \mathbb{R}^w$ and column vectors of $W_2^T(0)$ are $v_1, \ldots, v_d \in \mathbb{R}^w$. For $k = 1, 2, \ldots, d$, we define vector $v'_k$ as

$$v'_k = v_k - P_1 v_k. \tag{14}$$

We claim that $P_1 v_k$ is very small. By law of large numbers, $\|v_k\| \leq \sigma \sqrt{w} \log d$ with high probability.

$$\|P_1 v_k\| \leq \|\frac{1}{\sigma^2 w} W_1 W_1^T(0) v_k\| + O(\sqrt{\frac{d}{w}} \|v_1\|) \tag{15}$$

$$= \frac{1}{\sigma^2 w} \|\langle u_1, v_k \rangle u_1 + \ldots + \langle u_w, v_k \rangle u_w\| + O(\sigma \sqrt{d} \log d) \tag{16}$$

Notice that $\langle u_j, v_k \rangle u_j, \forall j$ is a family of independent and identically distributed random vectors. For each of these random vectors, all entries have zero mean. The variance of any one of the entries is given by

$$\mathbb{E}\left[\langle u_i, v_k \rangle^2 \langle u_j, e_\ell \rangle^2\right] = O(\sigma^6 w) \tag{17}$$

We conclude that for each $\ell$ we have, by CLT,

$$\frac{1}{\sigma^3 w^{\frac{1}{2}}} \frac{\langle u_1, v_k \rangle \langle u_1, e_\ell \rangle + \ldots + \langle u_w, v_k \rangle \langle u_w, e_\ell \rangle}{\sqrt{d}} \xrightarrow{(d)} N(0, 1) \tag{18}$$

In particular,

$$\mathbb{P}\{\max_\ell \sum_{i=1}^w |\langle u_i, v_k \rangle \langle u_k, e_\ell \rangle| > 100\sigma^3 \sqrt{d} w^{\frac{1}{2}} \log d\} \leq w\mathbb{P}\{\sum_{i=1}^w |\langle u_i, v_k \rangle \langle u_k, e_1 \rangle| > 100\sigma^3 \sqrt{d} w^{\frac{1}{2}} \log d\}$$

$$\leq 2w\mathbb{P}\{N(0, 1) > 100 \log d\}$$

$$\leq O(d^{-50})$$

Therefore with high probability, $\max_\ell \sum_{i=1}^w |\langle u_i, v_k \rangle \langle u_k, e_\ell \rangle| \leq 100\sigma^3 \sqrt{d} w^{\frac{1}{2}} \log d$. This implies that with high probability,

$$\|\langle u_1, v_k \rangle u_1 + \ldots + \langle u_w, v_k \rangle u_w\| \leq \sqrt{w} \sigma^3 \sqrt{d} w^{\frac{1}{2}} \log d \tag{19}$$

$$\|P_1 v_k\| \leq \sigma \sqrt{d} \log d \tag{20}$$

Now let $W_2'^T$ be the matrix with column vector $v'_1, \ldots, v'_w$ and let $P_2$ be the projection to the column space of $W_2'^T$. By construction $\|W_2 - W_2'\|_{\text{op}} \leq O(\sigma \sqrt{d} \log d)$, so $\|W_2^T W_2 - W_2'^T W_2'\|_{\text{op}} \leq O(\sigma^2 \sqrt{wd} \log d)$. Since the nonzero eigenvalues of $W_2^T W_2$ is at most $O(\sigma^2 \sqrt{wd})$ from 1, we know that the nonzero singular values of $W_2'^T W_2'$ is also at most $O(\sigma^2 \sqrt{wd})$ from 1. We conclude that $\|P_2 - W_2^T W_2\|_{\text{op}} \leq O(\sigma^2 \sqrt{wd} \log d)$. $\qquad\square$

*Proof of Proposition B.1.* : We have

$$W_1^T W_1 W_1^T W_1(t) = W_1^T W_2^T W_2 W_1 + W_1^T (W_1 W_1^T(t) - W_2^T W_2(t)) W_1$$

$$= A^T A + W_1^T (W_1 W_1^T(0) - W_2^T W_2(0)) W_1$$

From lemma B.3 we see that as positive semi-definite matrix, for every constant $\varepsilon > 0$,

$$0 \leq W_1 W_1^T(0) \leq (1 + \varepsilon)\sigma^2 w I \tag{21}$$

with high probability. Therefore

$$-(1 + \varepsilon)\sigma^2 w W_1^T W_1 \leq W_1^T (P_1(0) - P_2(0)) W_1 \leq (1 + \varepsilon)\sigma^2 w W_1^T W_1. \tag{22}$$

By moving terms around and corollary, we have

$$(W_1^T W_1)^2 - (1 + \varepsilon)\sigma^2 w W_1^T W_1 + (1 + \varepsilon)^2 \frac{\sigma^4 w^2}{4} I \tag{23}$$

$$\leq A^T A + \frac{(1 + \varepsilon)^2}{4} \sigma^4 w^2 I \tag{24}$$

$$\leq (W_1^T W_1)^2 + (1 + \varepsilon) W_1^T W_1 + (1 + \varepsilon)^2 \frac{\sigma^4 w^2}{4} I \tag{25}$$

Theorem V.1.9 of [12] states that the square-root function is operator monotone, which implies that if $A \geq B$ then $\sqrt{A} \geq \sqrt{B}$. Taking square-root, we have

$$W_1^T W_1 - (1 + \varepsilon)\frac{\sigma^2 w}{2} Id \leq \sqrt{A^T A + (1 + \varepsilon)^2 \frac{\sigma^4 w^2}{4}} \leq W_1^T W_1 + (1 + \varepsilon)\frac{\sigma^2 w}{2} \qquad (26)$$

$$\square$$

## B.2 Strong Bound

The weak bound does not provide useful information if $\|W_1^T W_1\|_{op} << \sigma^2 w$. For this reason we prove strong bound, which provide useful information if $\|W_1^T W_1\|_{op} << \sigma^2 w^{1+\square}$ for some constant $\square > 0$. Recall that the evolution of weight matrix in gradient descent is given by the following.

$$\frac{d}{dt}W_1(t) = \eta W_2^T \nabla C \qquad (27)$$

$$\frac{d}{dt}W_2(t) = \eta \nabla C W_1^T \qquad (28)$$

The goal of this section is to prove that

$$W_1^T W_1 \approx \sqrt{A^T A + \sigma^4 w^2 I} \qquad (29)$$

For simplicity of notations, we shall assume that $C_1 = W_1^T W_1$, $C_2 = W_2 W_2^T$, $\hat{C}_1 = \sqrt{A^T A + \sigma^4 w^2 I}$ and $C_2 = \sqrt{AA^T + \sigma^4 w^2 I}$. It is easy of see that $\hat{C}_1$ and $\hat{C}_2$ are invertible. Our main result for this section is the following proposition.

**Proposition B.4.** *For every cost $C$ we have*

$$\|W_1^T W_1 - \sqrt{A^T A + \sigma^4 w^2 I}\|_{op} \leq min\{O(\sigma^2 w), O\left(\sqrt{\frac{d}{w}}\|W_1^T W_1\|_{op}\right)\} \qquad (30)$$

*Proof of Lemma B.4 .* : We start from the equations:

$$W_1^T \left[(W_2^T W_2 - W_1 W_1^T)^2 - \sigma^4 w^2 I\right] W_1 = C_1^3 - A^T A C_1 - C_1 A^T A - \sigma^4 w^2 C_1 + A^T C_2 A$$
$$W_2 \left[(W_2^T W_2 - W_1 W_1^T)^2 - \sigma^4 w^2 I\right] W_2^T = C_2^3 - AA^T C_2 - C_2 AA^T - \sigma^4 w^2 C_2 + A C_1 A^T.$$

Our goal is to show that $C_1, C_2$ are close to the solution $\hat{C}_1 = \sqrt{A^T A + \sigma^4 w^2 I}, \hat{C}_2 = \sqrt{AA^T + \sigma^4 w^2 I}$ with

$$0 = \hat{C}_1^3 - A^T A \hat{C}_1 - \hat{C}_1 A^T A - \sigma^4 w^2 \hat{C}_1 + A^T \hat{C}_2 A$$
$$0 = \hat{C}_2^3 - AA^T \hat{C}_2 - \hat{C}_2 AA^T - \sigma^4 w^2 \hat{C}_2 + A\hat{C}_1 A^T.$$

Apriori, the cubic equation for $C_1$ and $C_2$ might have multiple solutions. We give an intuitive argument explaining why $\hat{C}_1$ and $\hat{C}_2$ are the correct solutions. By selecting a proper basis, we assume $A = diag(a_1, \ldots, a_d)$ is diagonal. Assume that $(W_2^T W_2 - W_1 W_1^T)^2 = \sigma^2 w I$. Also assume that $C_1$ and $C_2$ both commute with $A$. In this case the equations for $C_1$ and $C_2$ reduces to cubic equation for scalars. Let $\lambda_1, \ldots, \lambda_d$ be the eigenvalues of $C_1$. Solving the equations for scalars, we have $\lambda_i = 0$ or $\pm\sqrt{a_i^2 + \sigma^4 w^2}$. By lemma B.2 and lemma B.3, we have

$$W_1 W_1^T(t) - W_2^T W_2(t) = W_1 W_1^T(0) - W_2^T W_2(0) = \sigma^2 w P_1 - \sigma^2 w P_2 + o(\sigma^2 w).$$

Since $W_1 W_1^T$ is positive semi-definite, all its eigenvalues are non-negative, and thus

$$W_1 W_1^T(t) \geq (\sigma^2 w + o(\sigma^2 w)) P_1.$$

Since the top $w$ eigenvalues of $W_1^T W_1$ is the same as the top $w$ eigenvalues of $W_1 W_1^T$, we conclude that $\lambda_i \geq \sigma^2 w(1 + o(1))$. This forces $\lambda_i = \sqrt{a_i^2 + \sigma^4 w^2}$.

Since $C_1, C_2$ are not assumed to be aligned with $A$, we cannot reduce the matrix cubic equations into scalar cubic equations. The high level idea for proving $dC_i := C_i - \hat{C}_i$ is small is the inverse function theorem.

1. Step 1: show that LHS of the equations are small.

2. Step 2: reduce the RHS of the equations to a linear function of $dC_1$ and $dC_2$. The system of equations is thus reduced to

$$\begin{pmatrix} \text{small} \\ \text{small} \end{pmatrix} = \begin{pmatrix} * & * \\ * & * \end{pmatrix} \begin{pmatrix} v_i^T dC_1 \\ u_i^T dC_2 \end{pmatrix}.$$

The $*$ matrix is now the "Jacobian" matrix, and $u_i, v_i$ are left and right singular vectors of $A$.

3. Step 3: prove that the "Jacobian" matrix $\begin{pmatrix} * & * \\ * & * \end{pmatrix}$ is strictly positive definite, thus proving that $v_i^T dC_1$ and $u_i^T dC_2$ have small magnitude for all $i$.

For gradient flow, $W_2^T W_2 - W_1 W_1^T$ is preserved and there exists projections $P_1$ and $P_2$ such that $W_2^T W_2 - W_1 W_1^T = \sigma^2 w(P_2 - P_1) + O(\sigma^2 \sqrt{wd})$. Therefore for every unit vector $v$ we have

$$\| v^T W_1^T \left[ (W_2^T W_2 - W_1 W_1^T)^2 - \sigma^4 w^2 I \right] W_1 \| \leq \|C_1\|_{op} \sigma^4 d^{\frac{1}{2}} w^{\frac{3}{2}}.$$

Substracting the second pair of equations from the first pair and denoting $dC_i = C_i - \hat{C}$, we obtain:

$$\|C_1\|_{op} O(\sigma^4 w^2 \sqrt{\frac{d}{w}}) = C_1^3 - \hat{C}_1^3 - A^T A dC_1 - dC_1 A^T A - \sigma^4 w^2 dC_1 + A^T dC_2 A \qquad (31)$$

$$\|C_2\|_{op} O(\sigma^4 w^2 \sqrt{\frac{d}{w}}) = C_2^3 - \hat{C}_2^3 - A A^T dC_2 - dC_2 A A^T - \sigma^4 w^2 dC_2 + A dC_1 A^T. \qquad (32)$$

Now since

$$C_1^3 - \hat{C}_1^3 = \hat{C}_1^2 dC_1 + \hat{C}_1 dC_1 C_1 + dC_1 C_1^2,$$

we substitute the above relation to equation 31 and obtain

$$\|C_1\|_{op} O(\sigma^4 w^2 \sqrt{\frac{d}{w}}) = \left( \hat{C}_1^2 - A^T A \right) dC_1 + \hat{C}_1 dC_1 C_1 + dC_1 \left( C_1^2 - A^T A \right) - dC_1 + A^T dC_2 A \qquad (33)$$

$$= \hat{C}_1 dC_1 C_1 + dC_1 \left( C_1^2 - A^T A \right) + A^T dC_2 A, \qquad (34)$$

and similarly for equation 32. For any singular value $s_i$ of $A$, with left and right singular vectors $u_i, v_i$, we multiply equation 34 to the left by $v_i^T$, and divide both sides by $\sigma^2 w$, to obtain an equation for $v_i^T dC_1$. Similarly, we obtain an equation for $u_i^T dC_2$: for $v_i^T dC_1$ and $u_i^T dC_2$:

$$\|C_1\|_{op} O(\sigma^2 \sqrt{wd}) = v_i^T dC_1 \left( \sqrt{\left( \frac{s_i}{\sigma^2 w} \right)^2 + 1} C_1 + \frac{1}{\sigma^2 w}(C_1^2 - A^T A) \right) + \left( \frac{s_i}{\sigma^2 w} \right) u_i^T dC_2 A$$

$$\|C_2\|_{op} O(\sigma^2 \sqrt{wd}) = u_i^T dC_{2,i} \left( \sqrt{\left( \frac{s_i}{\sigma^2 w} \right)^2 + 1} C_2 + \frac{1}{\sigma^2 w}(C_2^2 - A A^T) \right) + \left( \frac{s_i}{\sigma^2 w} \right) v_i^T dC_1 A^T.$$

Notice that $C_1^2 - A^T A = W_1^T(W_1 W_1^T - W_2^T W_2)W_1 = \sigma^2 w W_1^T(P_1 - P_2)W_1 + \|C_1\|O(\sigma^2 \sqrt{wd})$. In the two equations above, by replacing $C_1^2 - A^T A$ with $\sigma^2 w W_1^T(P_1 - P_2)W_1$, we are making an error of at most $\|C_1\|_{op} \|dC_1\|_{op} O(\sqrt{\frac{d}{w}})$. From weak bound we know that $\|dC_1\|_{op} \leq O(\sigma^2 w)$. Therefore the error we made by making the approximation on the right hand side can be absorbed into left hand side.

To show that $\|v_i^T dC_1\|$ and $\|u_i^T dC_2\|$ are small, it suffices to show that the $(d_{in} + d_{out}) \times (d_{in} + d_{out})$ block matrix

$$\begin{pmatrix} \sqrt{\left( \frac{s_i}{\sigma^2 w} \right)^2 + 1} C_1 + W_1^T(P_1 - P_2)W_1 & \left( \frac{s_i}{\sigma^2 w} \right) A \\ \left( \frac{s_i}{\sigma^2 w} \right) A^T & \sqrt{\left( \frac{s_i}{\sigma^2 w} \right)^2 + 1} C_2 + W_2(P_2 - P_1)W_2^T \end{pmatrix}$$

is strictly positive definite. This matrix can be further simplified to

$$
\begin{pmatrix} W_1^T & 0 \\ 0 & W_2 \end{pmatrix} \begin{pmatrix} \sqrt{\left(\frac{s_i}{\sigma^2 w}\right)^2 + 1}I + P_1 - P_2 & \left(\frac{s_i}{\sigma^2 w}\right)I \\ \left(\frac{s_i}{\sigma^2 w}\right)I & \sqrt{\left(\frac{s_i}{\sigma^2 w}\right)^2 + 1}I + P_2 - P_1 \end{pmatrix} \begin{pmatrix} W_1 & 0 \\ 0 & W_2^T \end{pmatrix}
$$

The inner matrix can then be rewritten as $RR^T$ where $R$ is defined as

$$
R = \begin{pmatrix} \sqrt{\sqrt{\left(\frac{s_i}{\sigma^2 w}\right)^2 + 1} + 1}P_1 + \sqrt{\sqrt{\left(\frac{s_i}{\sigma^2 w}\right)^2 + 1} - 1}P_2 \\ \sqrt{\sqrt{\left(\frac{s_i}{\sigma^2 w}\right)^2 + 1} - 1}P_1 + \sqrt{\sqrt{\left(\frac{s_i}{\sigma^2 w}\right)^2 + 1} + 1}P_2 \end{pmatrix}.
$$

Let $Q = \begin{pmatrix} W_1^T & 0 \\ 0 & W_2 \end{pmatrix}$. The "Jacobian" matrix is then $QR(QR)^T$. As described in the strategy, we need to show that the singular values are strictly positive. The smallest nonzero singular value of $QR(QR)^T$ is the same as the smallest nonzero singular value of $(QR)^T QR$. We expand $(QR)^T QR$ as follows.

$$
(QR)^T QR
$$
$$
= \left( \sqrt{\left(\frac{s_i}{\sigma^2 w}\right)^2 + 1} + 1 \right) P_1 W_1 W_1^T P_1 + \left(\frac{s_i}{\sigma^2 w}\right) P_1 W_1 W_1^T P_2
$$
$$
+ \left(\frac{s_i}{\sigma^2 w}\right) P_2 W_1 W_1^T P_1 + \left( \sqrt{\left(\frac{s_i}{\sigma^2 w}\right)^2 + 1} - 1 \right) P_2 W_1 W_1^T P_2
$$
$$
+ \left( \sqrt{\left(\frac{s_i}{\sigma^2 w}\right)^2 + 1} + 1 \right) P_2 W_2^T W_2 P_2 + \left(\frac{s_i}{\sigma^2 w}\right) P_1 W_2^T W_2 P_2
$$
$$
+ \left(\frac{s_i}{\sigma^2 w}\right) P_2 W_2^T W_2 P_1 + \left( \sqrt{\left(\frac{s_i}{\sigma^2 w}\right)^2 + 1} - 1 \right) P_1 W_2^T W_2 P_1
$$
$$
= \left( \sqrt{\left(\frac{s_i}{\sigma^2 w}\right)^2 + 1} + 1 \right) (P_1 + P_2)\sigma^2 w
$$
$$
+ 2(\sqrt{\left(\frac{s_i}{\sigma^2 w}\right)^2 + 1} - \left(\frac{s_i}{\sigma^2 w}\right))(P_1 W_2^T W_2 P_1 + P_2 W_1 W_1^T P_2)
$$
$$
+ \left(\frac{s_i}{\sigma^2 w}\right) (W_1 W_1^T + W_2^T W_2 - \sigma^2 w P_1 - \sigma^2 w P_2) + O(\sigma^2 \sqrt{wd})
$$
$$
= \left( \sqrt{\left(\frac{s_i}{\sigma^2 w}\right)^2 + 1} + 1 - \left(\frac{s_i}{\sigma^2 w}\right) \right) (P_1 + P_2)\sigma^2 w
$$
$$
+ 2(\sqrt{\left(\frac{s_i}{\sigma^2 w}\right)^2 + 1} - \left(\frac{s_i}{\sigma^2 w}\right))(P_1 W_2^T W_2 P_1 + P_2 W_1 W_1^T P_2)
$$
$$
+ \left(\frac{s_i}{\sigma^2 w}\right) (W_1 W_1^T + W_2^T W_2) + O(\sigma^2 \sqrt{wd}).
$$

where we used the fact that

$$
W_1 W_1^T = P_1 + P_1 W_2^T W_2 P_1 + P_1 W_1 W_1^T P_2 + P_2 W_1 W_1^T P_1 + P_2 W_1 W_1^T P_2.
$$

The $(d_{in} + d_{out})$-th eigenvalue of the above is lower bounded by $\sigma^2 w\sqrt{\left(\frac{s_i}{\sigma^2 w}\right)^2 + 1} + \sigma^2 w - \sigma^2 w \left(\frac{s_i}{\sigma^2 w}\right) \geq \sigma^2 w$. We conclude that

$$
\|dC_1\|_{op} + \|dC_2\|_{op} \leq O(\sqrt{\frac{d}{w}})\|C_1\|_{op},
$$

$\square$

Compared to lemma B.3, lemma B.4 gives a tighter bound on $\|C_1 - \hat{C}_1\|_{\text{op}}$ when $\|C_1\|_{\text{op}} \leq \sigma^2 w \sqrt{\frac{d}{w}}$.

As suggested by an anonymous referee, it is possible to obtain the same approximated dynamics of $A_{\theta(t)}$ by imposing a non-homogeneous balance condition (in a different setup). Assume that $W_1 W_1^T - W_2^T W_2 = 2\sigma^2 wI$. Then

$$C_1^2 + 2\sigma^2 w C_1 - A^T A = 0;$$

$$C_2^2 - 2\sigma^2 w C_2 - AA^T = 0.$$

Therefore $C_1 = -\sigma^2 w + \sqrt{A^T A + \sigma^4 w^2}$ and $C_2 = \sigma^2 w + \sqrt{AA^T + \sigma^4 w^2}$. Substituting into Gradient Flow equation, we have

$$\frac{dA}{dt} = -\eta(\sqrt{AA^T + \sigma^4 w^2} \nabla C + \nabla C \sqrt{A^T A + \sigma^4 w^2}).$$

The advantage of the setup is that it significantly simplifies the proof. The The limitation of the setup is that $W_1 W_1^T - W_2^T W_2 \neq \sigma^4 w^2 I$ if the initial variance of entries of $W_1$ and $W_2$ are comparable. In this case, $W_1 W_1^T - W_2^T W_2$ will have $w$ positive singular values and $w$ negative singular values, and the absolute value of positive and negative singular values are comparable. In the setup of our problem, the variance of entries of $W_1$ and $W_2$ equal.

## C  Gradient Flow Dynamics of $A_t$ in Active Regime

### C.1  Saddle to Saddle Regime

Let $A_t$ have the following dynamics:

$$d^2 \frac{d}{dt} A_t = (A^* - A_t)\sqrt{A_t^T A_t + \sigma^4 w^2 I} + \sqrt{A_t A_t^T + \sigma^4 w^2 I}(A^* - A_t).$$

The goal of this section is to prove that the singular vectors of $A_t$ is well-aligned with the singular vectors of $A^*$, throughout the Saddle-to-Saddle regime. In the rest of the section, we will assume the dependence of $A_t$ on $t$ and use $A$ to represent $A_t$. If at initialization, $A_t$ commutes with $A^*$, then throughout the training, $A_t$ will always commute with $A^*$. In this section, we use a delicate stability argument to show that if $A_t$ almost commute with $A^*$ at the beginning of the Saddle to Saddle regime, then it will continue to be almost commutative with $A^*$ throughout the training process.

**Definition C.1.** *Define $P_1$ be the family of $d \times d$ matrices $A$ that satisfies the following conditions.*

- *$s_K \geq C\sigma^2 w$, $s_{K+1} \leq C'\sigma^2 w$ and $\frac{s_{K+1}}{s_K} \leq c < \frac{1}{2}$ for some d-independent constants c, C and C'.*

- *If $a_k > a_{k+1}$, then $s_k - s_{k+1} \geq cs_k$ for some d-independent constant c.*

*Define $P_1'$ to be the family of $w \times w$ matrix $A$ such that $s_k - s_{k+1} \geq \frac{c}{2}s_k$ if $a_{k+1} < a_k$, $\frac{s_{K+1}}{s_K} \leq \frac{3}{4}$ and $s_K \geq c\sigma^2 w$. Let $\gamma > 0$ be constant. Define $P_2(C, \gamma)$ be the family of $w \times w$ matrices $A$ satisfying the following conditions.*

- *(alignment of signals). Let $A = USV^T$. Define*

$$x = 4K - \sum_{k:signal} \text{tr}(U^T(k, k)U(k, k) + V^T(k, k)V(k, k) + 2U(k, k)V(k, k)^T).$$

*$P_2(C, \gamma)$ is the family of matrix $A$ such that $x \leq Cd^{-\gamma}$.*

**Theorem C.2.** *Assume that*

$$d^2 \frac{dA}{dt} = (A^* - A)\sqrt{A^T A + \sigma^4 w^2 I} + \sqrt{AA^T + \sigma^4 w^2 I}(A^* - A)$$

$$A(0) \in P_1 \cap P_2(C_4, \gamma)$$

*Let $T = O(d \log d)$. Then $\forall t \in [0, T]$, we have $A_t \in P_1 \cap P_2(C, min(1, \gamma))$.*

*Proof.* A simple result of the induction lemma C.3, lemma C.4, lemma C.6 and lemma C.7. □

We will use the following induction lemma to show that the singular vectors of $A_t$ are roughly aligned with $A^*$.

**Lemma C.3.** *Assume that $P_1$ and $P_2$ be families of increasing sets, and let $P_1' \supset P_1$. Assume that we have a family of matrices $A_t$, $0 \leq t \leq T$ for some fixed number $T$. $T$ does not depend on the family of matrices. Assume that $A_0 \in P_1 \cap P_2$. Let $A_{[t_1,t_2]} = \{A_t : t_1 \leq t \leq t_2\}$. Assume the following are true.*

1. *If $A_{[0,t]} \in P_1$ then there exists a constant $\varepsilon > 0$ independent of $A_t$ such that $A_{[0,t+\varepsilon]} \subset P_1'$.*

2. *Let $t_1 < t_2$. If $A_{[0,t_2]} \subset P_1'$ and $A_{[0,t_1]} \in P_2$, then $A_{[0,t_2]} \subset P_2$.*

3. *Let $t_1 < t_2$. If $A_{[0,t_2]} \subset P_2$ and $A_{[0,t_1]} \in P_1$ then $A_{[0,t_2]} \subset P_1$.*

*Then $A_t \in P_1$ and $A_t \in P_2$, $\forall 0 \leq t \leq T$.*

*Proof.* Since $A_0$ satisfies $P_1$, use condition 1 we have $A_{[0,\varepsilon]} \subset P_1'$. Using condition 2, we know that $A_{[0,\varepsilon]} \subset P_2$. By condition 3 we know that $A_{[0,\varepsilon]} \subset P_1$ and by condition 2, $A_{[0,\varepsilon]} \subset P_2$. Iterate the argument. □

**Lemma C.4.** *If $A_t \in P_1$, then there exists $\tau = \sigma^2 \sqrt{d} w d^{-1}$ such that $A_{[t,t+\tau]} \subset P_1'$.*

*Proof.* Let $0 \leq s \leq \tau$. Using the approximation in time $[t, s+t]$ we have

$$d^2 \frac{dS}{dt} = I \odot \left( U^T A^* V \sqrt{S^2 + \sigma^4 w^2} + \sqrt{S^2 + \sigma^4 w^2} U^T A^* V - 2S \sqrt{S^2 + \sigma^4 w^2} \right)$$

For $j = 1, 2, \ldots, K+1$,

$$d^2 \left| \frac{ds_j}{dt} \right| \leq Cd(s_j + \sigma^2 w)$$

for some constant $C$ independent of $w$ and $A_t$. The conclusion follows from Grönwall.

□

**Lemma C.5.** *Assume that*

$$d^2 \frac{dA}{dt} = (A^* - A + E) \sqrt{A^T A + \sigma^2 w I} + \sqrt{A A^T + \sigma^4 w^2 I} (A^* - A + E).$$

*Then for signal $k$,*

$$d^2 \frac{d}{dt} tr[U^T(k,k) U(k,k)] = tr[U^T(k,k) \sum_{j \neq k} U(k,j) D(j,k)]$$

*Here for $j \neq k$,*

$$D(j,k) = R(k,j) \odot C(j,k) - R(j,k) \odot C(k,j)^T$$

*with $C = U^T(A^* + E)V$, $R_{pq} = \frac{s_p \tilde{s}_p + s_p \tilde{s}_q}{s_p^2 - s_q^2}$ for $1 \leq p, q \leq w$, $p \neq q$, and $R(k,j) = R_{n_k:n_{k+1}, n_j:n_{j+1}}$ being the $k, j$-th block of $R$.*

*Proof.* Use SVD dervative.

$$\frac{dU}{dt} = U\left(F \odot \left[U^T \frac{dA}{dt} VS + SV^T \frac{dA^T}{dt} U\right]\right) \tag{35}$$

$$= U\left(F \odot \left[(U^T(A^* + E)V - S)\sqrt{S^2 + \sigma^4 w^2}S + \sqrt{S^2 + \sigma^4 w^2}(U^T(A + E)^* V - S)S\right]\right) \tag{36}$$

$$+ U\left(F \odot \left[\sqrt{S^2 + \sigma^4 w^2}S(V^T(A^* + E)^T U - S) + S(V^T(A^* + E)^T U - S)\sqrt{S^2 + \sigma^4 w^2}\right]\right) \tag{37}$$

$$= U\left(F \odot \left[U^T(A^* + E)V\sqrt{S^2 + \sigma^4 w^2}S + \sqrt{S^2 + \sigma^4 w^2}U^T(A^* + E)VS\right]\right) \tag{38}$$

$$+ U\left(F \odot \left[\sqrt{S^2 + \sigma^4 w^2}SV^T(A^* + E)^T U + SV^T(A^* + E)^T U\sqrt{S^2 + \sigma^4 w^2}\right]\right) \tag{39}$$

Let $D = F \odot \left[U^T \frac{dA}{dt} VS + SV^T \frac{dA^T}{dt} U\right]$. Since $F$ is anti-symmetric and the term in square bracket is symmetric, we know $D$ is anti-symmetric. Then $\frac{dU}{dt} = UD$. Let $\tilde{S} = \sqrt{S^2 + \sigma^4 w^2}$, $C = U^T(A^* + E)V$. As a result, $\forall 1 \leq p, q \leq w$,

$$\frac{dU_{pq}}{dt} = \sum_{r:r \neq q} U_{pr} \frac{1}{s_q^2 - s_r^2} [(s_q \tilde{s}_q + \tilde{s}_r s_q)C_{rq} + (\tilde{s}_r s_r + s_r \tilde{s}_k)C_{rq}]$$

Let $R_{pq} = \frac{s_p \tilde{s}_p + s_p \tilde{s}_q}{s_p^2 - s_q^2}$ if $p \neq q$ and $R_{pp} = 0$. Then

$$\frac{dU}{dt}(k, k) = \sum_{j:j \neq k} U(k, j)\left(R(k, j) \odot C(j, k) - R(j, k) \odot C(k, j)^T\right)$$

$$D(j, k) = R(k, j) \odot C(j, k) - R(j, k) \odot C(k, j)^T.$$

As a result, for $n_k \leq p, q < n_{k+1}$,

$$\frac{1}{2}\frac{d}{dt} tr[U^T(k, k)U(k, k)] = \sum_{p,q \in [n_k, n_{k+1})} U_{pq} \sum_r U_{pr} D_{rq}$$

$$= \sum_{p,q,r \in [n_k, n_{k+1})} U_{pq} U_{pr} D_{rq} + \sum_{p,q \in [n_k, n_{k+1})} U_{pq} \sum_{r \notin [n_k, n_{k+1})} U_{pr} D_{rq}$$

$$= \sum_{p,q \in [n_k, n_{k+1})} U_{pq} \sum_{r \notin [n_k, n_{k+1})} U_{pr} D_{rq}$$

$$= tr[U^T(k, k) \sum_{j \neq k} U(k, j)D(j, k)]$$

$\square$

**Lemma C.6.** *If $A_{[t_1, t_2]} \subset P_1$ and $A_{t_1} \in P_2(C_4, \gamma)$, then*

$$d^2 \frac{dx}{dt} \leq -cdx + O(\sqrt{dx}),$$

*where $\forall c = \inf_{t \in [t_1, t_2]} min(\frac{(s_k^* - s_j^*)(s_k + s_j)(\tilde{s}_k + \tilde{s}_j)}{s_k^2 - s_j^2}, \frac{(s_k - s_j)(s_k^* + s_j^*)(\tilde{s}_k + \tilde{s}_j)}{s_k^2 - s_j^2})$. In particular, $A_{[t_1, t_2]} \subset P_2(C, min(\gamma, 1))$ for some constant $C$. If $\gamma < 1$ then we can take $C = C_4$.*

*Proof.* Use previous lemma. Let

$$x = 4K - \sum_{k:\text{signal}} tr(U^T(k, k)U(k, k) + V^T(k, k)V(k, k) + 2U(k, k)V(k, k)^T).$$

$x$ measures the alignment of singular vectors of $A$ with singular vectors of $A^*$. From $UU^T = I$ we have

$$\sum_\ell U(k,\ell)U(k,\ell)^T = I$$

and for every $j \neq k$,

$$U(k,j)U(k,j)^T \leq I - U(k,k)U(k,k)^T \leq O(x)$$

In particular, $\|U(k,j)\|_{op} \leq O(\sqrt{x})$. We first estimate $C$. For all $j = 1,2,\ldots,w$, $a_j \neq a_k$,

$$
\begin{aligned}
(U^T A^* V)(j,k) =& U^T(j,j)S^*(j,j)V(j,k) + U^T(j,k)S^*(k,k)V(k,k) + \sum_{\ell \neq j,k} U^T(j,\ell)S^*_\ell V(\ell,k) \\
=& U^T(j,j)S^*(j,j)V(j,k) + U(k,j)^T S^*(k,k)V(k,k) \\
& + O(dx)
\end{aligned}
$$

$$
\begin{aligned}
(U^T E V)(j,k) =& \sum_{\ell_2 \neq k} U^T(j,j)E(j,\ell_2)V(\ell_2,k) + \sum_{\ell_1 \neq j} U^T(j,\ell_1)E(\ell_1,k)V(k,k) \\
& + U^T(j,j)E(j,k)V(k,k) + \sum_{\ell_1 \neq j,\ell_2 \neq k} U^T(j,\ell_1)E(\ell_1,\ell_2)V(\ell_2,k) \\
=& O(\sqrt{d}\sqrt{x}) + O(\sqrt{d}) + O(\sqrt{d}x) \\
=& O(\sqrt{d})
\end{aligned}
$$

$$
\begin{aligned}
D(j,k) =& R(k,j) \odot C(j,k) - R(j,k) \odot C(k,j)^T \\
=& R(k,j) \odot (U(j,j)^T S^*(j,j)V(j,k)) + R(k,j) \odot U(k,j)^T S^*(k,k)V(k,k) \\
& - R(j,k) \odot (V(k,j)^T S^*(k,k)U(k,k) + V(j,j)^T S^*(j,j)U(j,k)) \\
& + O(dx + \sqrt{d})
\end{aligned}
$$

By $UU^T = I$ we know that $\sum_\ell U(j,\ell)U^T(\ell,k) = 0$. Therefore

$$U(j,j)U(k,j)^T + U(j,k)U(k,k)^T + O(x) = 0$$

$$U(j,k) = -O(x) - U(j,j)U(k,j)^T U(k,k).$$

Similarly we have for $V$,

$$V(k,j) = -O(x) - V(k,k)V(j,k)^T V(j,j).$$

We can rewrite $D(j,k)$ as

$$
\begin{aligned}
D(j,k) =& R(k,j) \odot (U(j,j)^T S^*(j,j)V(j,k) + U(k,j)^T S^*(k,k)V(k,k)) \\
& + R(j,k) \odot (V(j,j)^T V(j,k)V(k,k)^T S^*(k,k)U(k,k) + V(j,j)^T S^*(j,j)U(j,j)U(k,j)^T U(k,k)) \\
& + O(dx + \sqrt{d})
\end{aligned}
$$

$$d^2 \frac{d}{dt} \text{tr} \left[ U(k,k)^T U(k,k) \right]$$

$$= 2\text{tr} \left[ U(k,k)^T \sum_j U(k,j) D(j,k) \right]$$

$$= 2 \sum_j \text{tr} \left[ U(k,k)^T U(k,j) \left( R(k,j) \odot (U(j,j)^T S^*(j,j) V(j,k) + U(k,j)^T S^*(k,k) V(k,k)) \right) \right]$$

$$+ 2 \sum_j \text{tr} \left[ U(k,k)^T U(k,j) \left( R(j,k) \odot (V(j,j)^T V(j,k) V(k,k)^T S^*(k,k) U(k,k)) \right) \right]$$

$$+ 2 \sum_j \text{tr} \left[ U(k,k)^T U(k,j) \left( R(j,k) \odot (V(j,j)^T S^*(j,j) U(j,j) U(k,j)^T U(k,k)) \right) \right]$$

$$+ O(dx^{\frac{3}{2}} + \sqrt{d}\sqrt{x})$$

$$\geq -2 \sum_j |S^*(j,j) R(k,j) + S^*(k,k) R(j,k)| \sqrt{tr(U(k,j)^T U(k,j))} \sqrt{tr(V(j,k)^T V(j,k))}$$

$$+ 2 \sum_j S^*(k,k) \text{tr}[U(k,k)^T U(k,j) R(k,j) \odot (U(k,j)^T U(k,k))]$$

$$+ 2 \sum_j S^*(j,j) \text{tr}[U(k,k)^T U(k,j) R(j,k) \odot (U(k,j)^T U(k,k))]$$

$$+ O(dx^{\frac{3}{2}}) + O(\sqrt{dx})$$

We know that

$$(S^*(k,k)R(k,j) + S^*(j,j)R(j,k)) - |S^*(j,j)R(k,j) + S^*(k,k)R(j,k)|$$
$$= \min\left( \frac{(s_k^* - s_j^*)(s_k + s_j)(\tilde{s}_k + \tilde{s}_j)}{s_k^2 - s_j^2}, \frac{(s_k - s_j)(s_k^* + s_j^*)(\tilde{s}_k + \tilde{s}_j)}{s_k^2 - s_j^2} \right)$$
$$\geq cd$$

for some positive constant $c$ independent of $w$. We conclude that

$$d^2 \frac{d}{dt} tr[U(k,k)^T U(k,k)] \geq cdx + O(dx^{\frac{3}{2}}) + O(\sqrt{dx}).$$

The same trick applies to $V$. We similarly have

$$d^2 \frac{d}{dt} tr[V(k,k)^T V(k,k)] \geq cdx + O(dx^{\frac{3}{2}}) + O(\sqrt{dx}).$$

It remains to show that $\frac{d}{dt} tr[V(k,k)^T U(k,k)]$ bounded below.

$$d^2 \frac{d}{dt} \text{tr}[U(k,k)^T V(k,k)] = tr[U(k,k)^T \frac{dV}{dt}(k,k)] + tr[V(k,k)^T \frac{dU}{dt}(k,k)]$$

$$tr[V(k,k)^T \frac{d}{dt} U(k,k)] = tr[U(k,k)^T \frac{d}{dt} U(k,k)] + tr[(V(k,k)^T - U(k,k)^T) \frac{d}{dt} U(k,k)]$$

$$\geq tr[U(k,k)^T \frac{d}{dt} U(k,k)] - \|V(k,k) - U(k,k)\|_F \| \frac{d}{dt} U(k,k) \|_F$$

$$\geq O(dx^{\frac{3}{2}}) + cdx + O(dx^{\frac{3}{2}}) + O(\sqrt{dx})$$

$$\geq cdx + O(dx^{\frac{3}{2}}) + O(\sqrt{dx})$$

Similarly,

$$tr[U(k,k)^T \frac{d}{dt} V(k,k)] \geq cdx + O(dx^{\frac{3}{2}}) + O(\sqrt{dx}).$$

This proves that

$$d^2 \frac{dx}{dt} \leq -cdx + O(dx^{\frac{3}{2}}) + O(\sqrt{dx}).\tag{40}$$

Observe that if $x \geq d^{-1+\varepsilon}$ for some $\varepsilon > 0$, then $\frac{dx}{dt} \leq -\frac{c}{2}dx$ and therefore $x$ decrease exponentially. On the other hand, if $x \leq d^{-1-\varepsilon}$ for some $\varepsilon > 0$, then $\frac{dx}{dt} \geq O(\sqrt{dx})$ and therefore $x$ might increase. We therefore conclude that $A_{[t_1,t_2]} \subset P_2(C, \min(\gamma, 1))$. $\qquad\square$

**Lemma C.7.** *Assume that* $A_{[0,t_2]} \subset P_2(C_4, \gamma)$ *and* $A_{[0,t_1]} \subset P_1$. *Then* $A_{[0,t_2]} \subset P_1$.

*Proof.* By assumption we know that $t_2 = O(d\log d)$.

$$d^2 \frac{dS}{dt} = I \odot \left( U^T(A^* + E)V\sqrt{S^2 + \sigma^4 w^2} + \sqrt{S^2 + \sigma^4 w^2}U^T(A^* + E)V - 2S\sqrt{S^2 + \sigma^4 w^2} \right)$$

$$d^2 \frac{ds_{K+1}}{dt} = 2\left( O(\sqrt{d}) + O(d^{1-\gamma}) - s_{K+1} \right)\sqrt{s_{K+1}^2 + \sigma^4 w^2}$$

$$|s_{K+1}(t_2) - s_{K+1}(t_1)| \leq O(\sigma^2 w(d^{-\frac{1}{2}} + d^{-\gamma})\log d) << \sigma^2 w$$

It remains to verify the gap between different families. Let $T$ be the first time when $\|A\|_{op} = \frac{1}{2}s_K^*$ and assume that $t_2 \leq T$. Assume that $a_k > a_j$, then there exists some constant $\varepsilon$ such that

$$d^2 \frac{ds_k}{dt} \geq (s_k^*(1 - \varepsilon) - s_k)s_k$$

$$d^2 \frac{ds_j}{dt} \leq (s_j^*(1 + \varepsilon) - s_j)(s_j + \sigma^2 w)$$

An application of Grönwall inequality on $s_k$ and $s_j$ implies that there exists some constant $\delta > 0$ that depends only on $a_j, a_k$ such that $\frac{s_j(T)}{s_k(T)} \leq O(d^{-\delta})$. Next we deal with the case when $t_2 > T$. One important observation is that the $\inf\{t > T : s_1(t) \geq \frac{a_k d + a_j d}{2}\} - T = O(d)$, as a simple result of Grönwall. Moreover, $s_2(T + O(d)) \leq \frac{1}{4}s_K^*$. This implies that $s_1 - s_k > cs_1$ for some constant $c$. Repeat this argument for all remaining signal singular values completes the proof. $\qquad\square$

## C.2 First NTK Regime

At initialization, the singular values of $A$ are of order $\sigma^2\sqrt{wd}$ by using Mechenko-Pastur law on $A^T A$, which is infinitely smaller than $\sigma^2 w$. As a result, there is a very short period when the dynamics is very close to NTK. This section is again a delicate stability argument to show that at the end of the first NTK regime, $A_t$ is almost commutative with $A^*$.

**Lemma C.8.** *Let* $t_1$ *be the first when the first singular value of* $A$ *hits* $\sigma^2 wd^{-\frac{\delta}{2}}$ *for* $\delta = \frac{\gamma_w - 1}{2}$. *Then* $A_{t_1} \in P_2(C, \min(\frac{1}{2}, \frac{\delta}{2}))$ *for some constant* $C$.

*Proof.* We approximate the dynamics with NTK dynamics. Assume that

$$d^2 \frac{dB}{dt} = 2(A^* + E - B)\sigma^2 w$$

with $B(0) = A(0)$. Then

$$d^2 \frac{d(A - B)}{dt} = 2(B - A)\sigma^2 w(A^* + E - A)$$
$$+ (\sqrt{A^T A + \sigma^4 w^2} - \sigma^2 w) + (\sqrt{AA^T + \sigma^4 w^2} - \sigma^2 w)(A^* + E - A).$$

A simple application of Grönwall inequality implies that

$$\|A - B\|_{op}(t) \leq \frac{1}{d^2}e^{-2\sigma^2 wt}\int_0^t e^{2\sigma^2 w\tau}\frac{2s_1(\tau)^2}{\sigma^2 w}\|A^* + E - A\|_{op}(\tau)d\tau$$

$$\leq 3\frac{\|A_t\|_{op}^2}{\sigma^2 w}s_1^*\frac{t}{d^2}$$

We first check $P_2$. Clearly, $t_1$ is of order $d^{1-\frac{\delta}{2}}$. Then we have

$$\|A_{t_1} - \frac{t_1}{d^2}\sigma^2 w A^*\|_{op} \leq 2d^{-\delta}\|A_{t_1}\|_{op} + \|\frac{t_1}{d^2}\sigma^2 w E\|_{op} + O(\sigma^2\sqrt{wd})$$

$$= (O(d^{-\delta}) + O(d^{-\frac{1}{2}}) + O(d^{-\frac{\delta}{2}}))\|A_{t_1}\|_{op}$$

By lemma C.4 we have $x \leq Cd^{-\min(\frac{1}{2},\frac{\delta}{2})}$ for some constant $C$. $\qquad\square$

**Lemma C.9.** *Let $t_2 = cd$, where $c$ is a constant to be chosen. Then*

$$A_{t_2} \in P_1 \cap P_2(C, min(\frac{\gamma_w - 1}{4}, \frac{1}{2})).$$

*Moreover, there exists constants $b_1 = \ldots b_{n_1} > b_{n_1+1} = \ldots = b_{n_2} \geq \ldots = b_{n_m} = b_K$, such that for a $d \times d$ diagonal matrix $\Sigma$ whose diagonal entries are given by $b_1, \ldots, b_K, 0, \ldots, 0$, we have $\|A_{t_2} - \Sigma\|_{op} \leq O(\sigma^2 w)d^{-min(\frac{\gamma_w-1}{8},\frac{1}{4})}.$*

*Proof.* At time $t_2$, $\|B(t_2)\|_{op} = 2a_1 c\sigma^2 w(1 + o(1))$. From previous lemma we know that

$$\|A - B\|_{op}(t) \leq 3\frac{\|A_t\|_{op}^2}{\sigma^2 w}s_1^* \frac{t}{d^2}.$$

Let $t = t_2$ we see that $|\|A(t_2)\|_{op} - \|B(t_2)\|_{op}| \leq \|A - B\|_{op}(t_2) \leq 2\frac{\|A_{t_2}\|_{op}^2}{\sigma^2 w}a_1 c$. Therefore as long as $c$ is sufficiently small, we can guarantee that $\|A - B\|_{op}(t) \leq \frac{\min_{1 \leq k \leq K}(a_{k-1}-a_k)}{100a_1}\|B\|_{op}(t)$. This guarantees that the singular values of $A$ grows linearly in $[t_1, t_2]$, that $s_k - s_{k+1} \geq cs_k$ if $a_k > a_{k+1}$ for some constant $c$, that $s_{K+1} < \frac{1}{2}s_K$ and that $A_{t_2} \in P_1$. It remains to check that $A_{t_2}$ satisfies $P_2(C, \frac{\gamma_w-1}{2})$. We are ready to use similar techniques as in lemma C.6. All computation are similar, and the only difference is that $|R(k, j)| = O(\frac{\sigma^2 w}{s_K})$. We have

$$d^2\frac{d}{dt}\text{tr}[U(k,k)^T U(k,k)] \geq c\frac{\sigma^2 w}{s_K}dx + O(dx^{\frac{3}{2}}\frac{\sigma^2 w}{s_K}) + O(\sqrt{dx}\frac{\sigma^2 w}{s_K})$$

$$d^2\frac{dx}{dt} \leq 2C\frac{\sigma^2 w}{a_K(\sigma^2 wwt + \sigma^2 wd^{-\frac{\delta}{2}})}(-cdx + dx^{\frac{3}{2}} + \sqrt{dx})$$

Therefore $A_{[t_1,t_2]} \subset P_2(C, \min(\frac{\gamma_w-1}{4}, \frac{1}{2}))$. Let $\gamma = \min(\frac{\gamma_w-1}{4}, \frac{1}{2})$. From lemma C.8 we see that if $n_{k_1} + 1 \leq p_1, p_2 \leq n_k$, then $|s_{p_1} - s_{p_2}| \leq O(d^{-\gamma})s_K$. Moreover, the dynamics of $s_{p_i}, i = 1, 2$ are both given by

$$d^2\frac{ds_{p_i}}{dt} = 2(O(\sqrt{d}) + O(d^{1-\gamma}) + a_k d - s_{p_i})\sqrt{s_{p_i}^2 + \sigma^4 w^2}.$$

$$\frac{d|s_{p_1} - s_{p_2}|}{dt} \leq O(d^{-1})|s_{p_1} - s_{p_2}|$$

$$|s_{p_1} - s_{p_2}|(t_2) \leq O(1)|s_{p_1} - s_{p_2}|(t_1) = \sigma^2 wO(d^{-\gamma-\frac{\gamma_w-1}{4}}).$$

Let $b_{n_i} = s_i$. By lemma A.5, there exists a $d \times d$ diagonal matrix $\Sigma$, whose diagonal entries are given by $b_1 = \ldots = b_{n_1} > \ldots = b_{n_2} \geq \ldots = b_K$, such that $\|A_{t_2} - \Sigma\|_{op} \leq O(\sigma^2 w)d^{-\frac{\gamma}{2}}.$ $\qquad\square$

# D   The Gradient Flow Dynamics of $A_{\theta(t)}$ in Lazy Regime

In this section, we use a stability argument to show that if $\sigma^2 w$ is infinitely larger than $d$, then the algorithm cannot converge.

**Proposition D.1.** *With high probability, for all time $t$, $\|A_{\theta(t)} - A^*\|_F^2 \geq \frac{1}{3}min(\|A^*\|_F^2, \|E\|_F^2).$*

*Proof.* In gradient flow dynamics, $\|A_{\theta(t)} - A^* - E\|_F$ is decreasing with time and therefore for all time $t$, $\|A_{\theta(t)}\|_F^2 \leq 9(\|A^*\|_F^2 + \|E\|_F^2)$. In particular, $\|A_{\theta(t)}\|_{op} \leq 9(\|A^*\|_F + \|E\|_F)$. By

assumption, $\|\hat{C}_1 - C_1\|_{op} \leq O(\sqrt{\frac{d}{w}})\|C_1\|_{op} = O(d\sqrt{\frac{d}{w}})$, and the dynamics of $A_{\theta(t)}$ can be written as

$$d^2 \frac{dA_{\theta(t)}}{dt} = 2(A^* + E - A)\sigma^2 w + (A^* + E - A)O(d\sqrt{\frac{d}{w}}) + O(d\sqrt{\frac{d}{w}})(A^* + E - A)$$

Assume that $B(0) = A_{\theta(0)}$ and

$$d^2 \frac{dB}{dt} = 2(A^* + E - B)\sigma^2 w.$$

Then

$$\frac{1}{2}d^2 \frac{d}{dt}\|A_{\theta(t)} - B\|_F^2 = -2\sigma^2 w\|A_{\theta(t)} - B\|_F^2 + tr((A - B)^T(A^* + E - A)O(d\sqrt{\frac{w}{d}}))$$

$$+ tr((A - B)^T O(d\sqrt{\frac{w}{d}})(A^* + E - A))$$

$$\leq -2\sigma^2 w\|A_{\theta(t)} - B\|_F^2 + \|A_{\theta(t)} - B\|_F O(d^2\sqrt{\frac{d}{w}})$$

This implies that for all time,

$$\|A_{\theta(t)} - B\|_F \leq dO(\frac{d}{\sigma^2 w}\sqrt{\frac{d}{w}}).$$

The dynamics of $B$ is linear, and we have

$$B_t = (A^* + E)(1 - e^{-2\sigma^2 wt/d^2}) + B_0 e^{-2\sigma^2 wt/d^2}.$$

$$\|B_t - A^*\|_F^2 = \|e^{-2\sigma^2 t/d^2}A^* + (1 - e^{-2\sigma^2 wt/d^2})E + B_0 e^{-2\sigma^2 wt/d^2}\|_F^2 \geq 0.99\|e^{-2\sigma^2 t/d^2}A^* + (1 - e^{-2\sigma^2 wt/d^2})E\|_F^2$$

Let $P$ be the projection to the image of $A^*$. Then with high probability,

$$\|B_t - A^*\|_F^2 = \|e^{-2\sigma^2 wt/d^2}A^* + (1 - e^{-2\sigma^2 wt/d^2})E + B_0 e^{-2\sigma^2 wt/d^2}\|_F^2$$

$$\geq 0.99\|e^{-2\sigma^2 t/d^2}A^* + (1 - e^{-2\sigma^2 wt/d^2})E\|_F^2$$

$$= 0.99\|e^{-2\sigma^2 wt/d^2}A^* + (1 - e^{-2\sigma^2 wt/d^2})PE\|_F^2 + 0.99\|(1 - e^{-2\sigma^2 wt/d^2})(I - P)E\|_F^2$$

$$\geq \frac{1}{2}\min(\|A^*\|_F^2, \|E\|_F^2)$$

Since $\|A_{\theta(t)} - B\|_F^2 = o(d^2)$, we have

$$\|A_{\theta(t)} - A^*\|_F^2 \geq \frac{1}{3}\min(\|A^*\|_F^2, \|E\|_F^2)$$

$\square$

# E   The Gradient Flow Dynamics of $A_{\theta(t)}$ in Active Regime

In section C, we proved stability for gradient flow dynamics for $A_t$. In this section, we prove similar stability statements for $A_{\theta(t)}$, using the approximation results from section B. We also summarize the behavior of $A_{\theta(t)}$ in section E.3.

## E.1   Saddle-to-Saddle Regime

The goal of this section is to show that at the end of the first NTK regime, the singular vectors of $A_{\theta(t)}$ are roughly aligned with $A^*$. Moreover, the alignment remains to be good throughout the mixed regime. The following generalization of lemma C.6 and lemma C.7 will be useful.

**Lemma E.1.** *Assume that $A'_{[t_1,t_2]} \subset P_1$ and $A_{t_1} \in P_2(C_4, \gamma)$. Let $s'_1, \ldots, s'_K$ be the first $K$ singular values of $A'$. Assume that the dynamics of $A'$ is the following.*

$$d^2 \frac{dA'}{dt} = (A^* + E - A')\sqrt{A'^T A' + \sigma^4 w^2 I} + \sqrt{A'A'^T + \sigma^4 w^2}(A^* + E - A')$$
$$+ O(d^{-\lambda} s'_K)(A^* + E - A') + (A^* + E - A')O(d^{-\lambda} s'_K)$$

*Here, $O(d^{-\lambda} s'_K)$ is a matrix whose operator norm is bounded by $O(d^{-\lambda} s'_K)$. Then $A_{[t_1,t_2]} \subset P_2(C_4, \min(\gamma, 1, 2\lambda))$.*

*Proof.* Let $A' = USV^T$. Computing the SVD derivative for $A'$, we have

$$d^2 \frac{dU}{dt} = U\left(F \odot \left[U^T A^* V \sqrt{S^2 + \sigma^4 w^2} S + \sqrt{S^2 + \sigma^4 w^2} U^T A^* V S\right]\right)$$
$$+ U\left(F \odot \left[\sqrt{S^2 + \sigma^4 w^2} S V^T A^* U + V^T A^* U S \sqrt{S^2 + \sigma^4 w^2}\right]\right)$$
$$+ U\left(F \odot \left[O(s_K d^{-\lambda})(U^T A^* V S + S V^T A^* U - 2S^2)\right]\right)$$

Assume that $a_k \neq a_\ell$. Notice that

$$F_{k\ell}(O(s_k d^{-\lambda}) U^T A^* V S)_{k\ell} = O(d^{1-\lambda})$$

Using the same technique as C.6 we have

$$\frac{1}{2} d^2 \frac{d}{dt} tr[U(k,k)^T U(k,k)]$$
$$\geq cdx + O(dx^{\frac{3}{2}}) + O(\sqrt{dx})$$
$$+ tr[U(k,k)^T \sum_{j \neq k} U(k,j)(F \odot \left[O(s_K d^{-\lambda})(U^T A^* V S + S V^T A^* U - 2S^2)\right])]$$
$$\geq cdx + O(dx^{\frac{3}{2}}) + O(d^{1-\lambda}\sqrt{x}) + O(\sqrt{dx})$$

Similar conclusion also holds for $tr[U(k,k)^T V(k,k)]$ and $tr[V(k,k)^T V(k,k)]$. We conclude that

$$d^2 \frac{dx}{dt} \leq -cdx + O(dx^{\frac{3}{2}}) + O(d^{1-\lambda}\sqrt{x}) + O(\sqrt{dx}),$$

and $A'_{[t_1,t_2]} \subset P_2(C_4, \min(\gamma, 1, 2\lambda))$. $\square$

**Lemma E.2.** *Assume that $A'_{[t_1,t_2]} \subset P_2(C_4, \gamma)$ and $A_{t_1} \in P_1$. Let $s'_1, \ldots, s'_K$ be the first $K$ singular values of $A'$. Assume that the dynamics of $A'$ is the following.*

$$d^2 \frac{dA'}{dt} = (A^* - A')\sqrt{A'^T A' + \sigma^4 w^2 I} + \sqrt{A'A'^T + \sigma^4 w^2}(A^* - A') + O(d^{-\lambda} s'_K)(A^* - A) + (A^* - A)O(d^{-\lambda} s'_K)$$

*Here, $O(d^{-\lambda} s'_K)$ is a matrix whose operator norm is bounded by $O(d^{-\lambda} s'_K)$. Then $A_{[t_1,t_2]} \subset P_1$.*

*Proof.* Proceeding as in lemma C.7, we have

$$d^2 \frac{ds_p}{dt} = 2(O(\sqrt{d}) + O(d^{1-\gamma}) + a_p d - s_p)(\sqrt{s_p^2 + \sigma^4 w^2} + O(d^{-\lambda}) s_K)$$

if $p$ is a signal, and

$$d^2 \frac{ds_{K+1}}{dt} = 2(O(\sqrt{d}) + O(d^{1-\gamma}) - s_{K+1})(\sqrt{s_{K+1}^2 + \sigma^4 w^2} + O(d^{-\gamma}) s_K).$$

Now the conclusion follows from Grönwall. $\square$

**Theorem E.3.** *Assume that $A_{\theta(t)} \subset P_1 \cap P_2(C, \gamma)$. Then $A_{[\theta([t,T])} \subset P_1 \cap P_2(C, \gamma')$ for some constant $\gamma'$.*

*Proof.* We use induction lemma C.3 to prove the theorem. For the first requirement, notice that for every $i = 1, 2, \ldots, w$,

$$d^2 \left| \frac{ds_i}{dt} \right| \leq 4\|A^*\|_{op}^2.$$

If $A_{\theta(t)} \in P_1$, then $A_{\theta([t,t+\tau])} \subset P_1'$ for $\tau = c\frac{\sigma^2\sqrt{wd}}{d^2\|A^*\|_{op}^2}$. Here $c$ is some small constant that depends only on $P_1$. This proves the first requirement. For the second requirement, we observe that $A_{\theta(t)}$ satisfies the dynamics described in lemma E.1 at each stage, as long as $\|\hat{C}_1 - C_1\|_{op} \leq O(s_K d^{-\lambda}$ for some $\lambda > 0$. It suffices to show that there exists some positive $\lambda$ independent of $w$ such that $\|\hat{C}_1 - C_1\|_{op} \leq O(d^{-\lambda}s_K)$ in $[0, T]$. Let $T_1$ be the first time when $\|A\|_{op} = \sigma^2 w$. In $[0, T_1]$, all singular values are of the same order. Therefore

$$\|\hat{C}_1 - C_1\|_{op} \leq O(\|C_1\|\sqrt{\frac{d}{w}}) = O(s_K d^{-\frac{\gamma_w - 1}{2}}).$$

Now let $T_2$ be the first time when $s_1 = \sigma^2 w d^{\frac{\gamma_w - 1}{4}}$. In $[T_1, T_2]$, we have

$$\|\hat{C}_1 - C_1\|_{op} \leq O(\|C_1\|\sqrt{\frac{d}{w}}) \leq O(s_K d^{-\frac{\gamma_w - 1}{4}})$$

Therefore we can pick $\lambda = \frac{\gamma_w}{4}$. During $[T_1, T_2]$, we have $s_K(0) \geq c\sigma^2 w$ for some constant $c$, and

$$d^2 \frac{ds_K}{dt} \geq s_K(\frac{1}{2}a_K d - s_K)$$

Moreover, since $d^2 \frac{ds_1}{dt} \leq (2a_1 - s_1)(s_1 + \sigma^2 w)$, we have $T_2 - T_1 \geq c\frac{\log w}{w}$ for some $c$. Therefore we have $s_K(T_2) \geq \sigma^2 w d^{-\nu}$ for some constant $\nu > 0$. Now in $[T_2, T]$, we have weak bound $\|\hat{C}_1 - \hat{C}\|_{op} \leq O(\sigma^2 w)$. We can therefore pick $\lambda = \nu$ for $[T_2, T]$. The third requirement is verified in lemma E.2. □

## E.2 First NTK Regime

As in $A_t$, we need to show that at the end of the NTK regime, $A_{\theta(t)}$ must be in $P_1$ and $P_2$. Based on what we already have for $A_t$, this conclusion follows from Grönwall.

**Lemma E.4.** *Assume that $A_0 = A_{\theta(0)}$. Let $T_1'$ be the first time when $\|A_t\|_{op} + \|A_{\theta(t)}\|_{op}$ reaches $\sigma^2 w$. Then*

$$\|A_{T_1'} - A_{\theta(T_1')}\|_{op} \leq O(\sigma^2\sqrt{wd}).$$

*In particular, $A_{\theta(T_1')} \in P_1 \cap P_2(C, min(\frac{\gamma_w - 1}{8}, \frac{1}{4}))$.*

*Proof.* From dynamics of $A_t$ we see that $T_1' \leq \frac{c}{w}$ for some constant $c$. The dynamics of $A_t$ is the following.

$$d^2 \frac{dA_t}{dt} = (A^* - A_t)\sqrt{A_t^T A_t + \sigma^2 wI} + \sqrt{A_t A_t^T + \sigma^4 w^2 I}(A^* - A_t).$$

The dynamics of $A_{\theta(t)}$ can be written as follows.

$$d^2 \frac{dA_{\theta(t)}}{dt} = (A^* - A_{\theta(t)})\sqrt{A_{\theta(t)}^T A_{\theta(t)} + \sigma^4 w^2 I} + \sqrt{A_{\theta(t)} A_{\theta(t)}^T + \sigma^4 w^2 I}(A^* - A_{\theta(t)})$$
$$+ O(\sigma^2\sqrt{wd})(A^* - A_{\theta(t)}) + (A^* - A_{\theta(t)})O(\sigma^2\sqrt{wd})$$

Observe that

$$\|A_{\theta(t)}^T A_{\theta(t)} - A_t^T A_t\|_{op} \leq \|A_{\theta(t)}^T(A_{\theta(t)} - A_t)\|_{op} + \|(A_{\theta(t)} - A_t)^T A_t\|_{op}$$
$$\leq \sigma^2 w\|A_{\theta(t)} - A_t\|_{op}.$$

This implies that we have the following inequality on positive definite matrices.

$$A_t^T A_t - \sigma^2 w\|A_{\theta(t)} - A_t\|_{op}I \leq A_{\theta(t)}^T A_{\theta(t)} \leq A_t^T A_t + \sigma^2 w\|A_{\theta(t)} - A_t\|_{op}I$$

Assume that $A_t = USV^T$, then

$$\|\sqrt{A_{\theta(t)}^T A_{\theta(t)} + \sigma^4 w^2 I} - \sqrt{A_t^T A + \sigma^4 w^2 I}\|_{op}$$

$$\leq \|\sqrt{\sigma^4 w^2 I + A_t^T A_t + \sigma^2 w \|A_{\theta(t)} - A_t\|_{op} I} - \sqrt{\sigma^4 w^2 I + A_t^T A_t - \sigma^2 w \|A_{\theta(t)} - A_t\|_{op} I}\|_{op}$$

$$= \|\sqrt{\sigma^4 w^2 I + \sigma^2 w \|A_{\theta(t)} - A_t\|_{op} I + S^2} - \sqrt{\sigma^4 w^2 I - \sigma^2 w \|A_{\theta(t)} - A_t\|_{op} I + S^2}\|_{op}$$

$$= \max_i \left( \sqrt{\sigma^4 w^2 + s_i^2 + \sigma^2 w \|A_{\theta(t)} - A_t\|_{op}} - \sqrt{\sigma^4 w^2 + s_i^2 - \sigma^2 w \|A_{\theta(t)} - A_t\|_{op}} \right)$$

$$\leq 3\|A_{\theta(t)} - A_t\|_{op}$$

We can control the difference between $A_t$ and $A_{\theta(t)}$.

$$d^2 \frac{d}{dt} \|A_t - A_{\theta(t)}\|_{op} \leq O(d)\|A_t - A_{\theta(t)}\| + O(d)O(\sigma^2 \sqrt{wd})$$

Grönwall inequality implies that

$$\|A_{T_1'} - A_{\theta(T_1')}\|_{op} \leq O(\sigma^2 \sqrt{wd})$$

We check that $A_{\theta(T_1')}$ satisfies the $P_1$. Since $\|A_{T_1'} - A_{\theta(T_1')}\|_{op} \leq O(\sigma^2 \sqrt{wd})$, $\|A_{\theta(T_1')}\|_{op} \geq \frac{1}{3}\sigma^2 w$. Let $\sigma_j(A_{T_1'})$ and $\sigma_j(A_{\theta(T_1')})$ be the $j$-th singular value of $A_{T_1'}$ and $A_{\theta(T_1')}$. If $a_j \neq a_k$, $j, k \leq K$, then $|\sigma_j(A_{T_1'}) - \sigma_k(A_{T_1'})| \geq c\sigma^2 w$, and therefore

$$|\sigma_j(A_{\theta(T_1')}) - \sigma_k(A_{\theta(T_1')})| \geq c\sigma^2 w + O(\sigma^2 \sqrt{wd}).$$

The noise is clearly of order $O(\sigma^2 \sqrt{wd})$. This completes the verification of $P_1$. By lemma C.9, $\|A_{\theta(T_1')} - \Sigma\|_{op} \leq \sigma^2 w O(\sigma^2 w) d^{-\min(\frac{\gamma_w - 1}{8}, \frac{1}{4})}$. By Lemma C.4 we are done.

$\square$

### E.3  Summary of Approximate Dynamics of $A_{\theta(t)}$ at Each Stage

In previous sections we have proved that $A_{\theta(t)}$ satisfies $P_1$ and $P_2(C, \gamma)$ for some $\gamma > 0$. The $P_1$ and $P_2(C, \gamma)$ property actually implies that the dynamics of $A_{\theta(t)}$ is such that each group of singular values evolve independently. We state the approximate dynamics for each stage of the dynamics, and show that the alignment will be improved, if the alignment is not already good enough.

- **Initialization**. At initialization, $A_{\theta(0)}$ is a random matrix. The mean of each entry is 0 and the variance of each entry is $\sigma^2 \sqrt{w}$. The gap between singular values is infinitely smaller than the magnitude of singular values, and the singular vectors are not aligned with $A^*$.

- **Initialization to** $\|A_{\theta(t)}\|_{op} = \sigma^2 w$. Let $T_1$ be the first time when $\|A_{\theta(t)}\|_{op}$ reaches $\sigma^2 w$. $[0, T_1]$ corresponds the very short NTK regime for the signals, and the dynamics of $A_{\theta(t)}$ is approximately linear. The evolution of signal singular values are roughly linear (i.e., bounded above and below by linear functions), and at $T_1$, we have $A_{\theta(T_1)} \in P_2(C, \min(\frac{\gamma_w - 1}{8}, \frac{1}{4}))$. Since the singular values grows roughly linearly, $T_1 = O(d)$.

- $\|A_{\theta(t)}\|_{op} = \sigma^2 w$ **to** $\|A_{\theta(t)}\|_{op} = \sigma^2 w \left(\frac{w}{d}\right)^{\frac{1}{4}}$. Let $T_2$ be the first time when $\|A_{\theta(t)}\|_{op} = \sigma^2 w \left(\frac{w}{d}\right)^{\frac{1}{4}}$. $[T_1, T_2]$ is the early stage of saddle-to-saddle dynamics. The dynamics of $s_i$ is given by

$$d^2 \frac{ds_i}{dt} = 2(a_i d(1 + o(1)) - s_i)\sqrt{s_i^2 + \sigma^4 w^2}(1 + o(1)).$$

By theorem 1, we have $\|\hat{C}_1 - C_1\|_{op} \leq \sigma^2 w \left(\frac{w}{d}\right)^{-\frac{1}{4}}$. Let $d^{-\lambda}\sigma^w = \sigma^2 w \left(\frac{w}{d}\right)^{-\frac{1}{4}}$, we have $\lambda = \frac{\gamma_w - 1}{4}$. By lemma 4,1, the dynamics of $x$ satisfies

$$d^2 \frac{dx}{dt} \leq -cdx + O(dx^{\frac{3}{2}}) + O(\sqrt{x}d^{1 - \frac{\gamma_w - 1}{4}}) + O(\sqrt{dx})$$

with $x(T_1) = \frac{\gamma_w - 1}{2}$. We conclude that

$$x(T_2) \leq O(d^{-\frac{\gamma_w - 1}{2}}).$$

- $\|A_{\theta(t)}\|_{op} = \sigma^2 w \left(\frac{w}{d}\right)^{\frac{1}{4}}$ to $s_K(A_{\theta(t)}) = \frac{1}{2}a_K d$. Let $T_3$ be the first time $s_K(A_{\theta(t)}) = \frac{1}{2}a_K d$. At time $T_2$, we have $s_K \geq \sigma^2 w d^\delta$ for some $\delta > 0$ that depends only on $a_1, \ldots, a_K$. Then we have

$$\|\hat{C}_1 - C_1\|_{op} \leq O(s_K d^{-\delta}),$$

$$d^2 \frac{dx}{dt} \leq -cdx + O(dx^{\frac{3}{2}}) + O(\sqrt{x}d^{1-\delta}) + O(\sqrt{dx}).$$

We conclude that $x(T_3) = O(d^{-2\delta})$. The dynamics of $s_i$ is given by

$$d^2 \frac{ds_i}{dt} = 2(a_i d(1 + O(d^{-2\delta})) - s_i)\sqrt{s_i^2 + \sigma^4 w^2}(1 + O(d^{-2\delta})).$$

$$s_i(a_i d - s_i) \leq d^2 \frac{ds_i}{dt} \leq 2(a_i d + Cd^{1-2\delta} - s_i)(s_i + \sigma^2 w).$$

The bounds on $\frac{ds_i}{dt}$ implies that

$$T_3 - T_1 = \frac{d \log \frac{a_K d}{\sigma^2 w}}{a_K} + O(d) = \frac{1 - \gamma_{\sigma^2} - \gamma_w}{a_K} d \log d + O(d).$$

- **Final Stage**. Let $t \geq T_3$. Since $s_K = O(d)$, we have

$$\|\hat{C}_1 - C_1\|_{op} \leq O(s_K \frac{\sigma^2 w}{d}) = O(s_K d^{\gamma_{\sigma^2}+\gamma_w-1});$$

$$d^2 \frac{dx}{dt} \leq -cdx + O(dx^{\frac{3}{2}}) + O(\sqrt{x}d^{\gamma_{\sigma^2}+\gamma_w}) + O(\sqrt{dx}).$$

Recall that the constant $c$ in term $-cdx$ must satisfy

$$c \leq \frac{1}{d}\min_{k,j:a_k \neq a_j}\left(\frac{(s_k^* - s_j^*)(s_k + s_j)(\tilde{s}_k + \tilde{s}_j)}{s_k^2 - s_j^2}, \frac{(s_k - s_j)(s_k^* + s_j^*)(\tilde{s}_k + \tilde{s}_j)}{s_k^2 - s_j^2}\right)$$

As a result, we can take

$$c = c(a_1, \ldots, a_K) = \frac{\min_{k,j:a_k \neq a_j}|a_k - a_j|a_K^2}{\max_{k,j:a_k \neq a_j}|a_k^2 - a_j^2|}$$

Let $c'$ be a large constant such that if $x > c'(d^{-1} + d^{2(\gamma_{\sigma^2}+\gamma_w-1)})$, then $\frac{dx}{dt} \leq -\frac{c(a_1,\ldots,a_K)d}{2}x$. Let $T_4$ be the first time when $x \leq c'(d^{-1} + d^{2(\gamma_{\sigma^2}+\gamma_w-1)})$ after $T_3$. Then $T_4 - T_3 \leq -\frac{2d}{c(a_1,\ldots,a_K)}\log\left(d^{-1} + d^{2(\gamma_{\sigma^2}+\gamma_w-1)}\right) + O(d)$. Moreover, for every $t > T_4$, $x$ cannot be larger than $c'(d^{-1} + d^{2(\gamma_{\sigma^2}+\gamma_w-1)})$ because $\frac{dx}{dt} < 0$ if $x = c'(d^{-1} + d^{2(\gamma_{\sigma^2}+\gamma_w-1)})$. After $T_4$, the dynamics of each singular value is given by

$$d^2 \frac{ds_i}{dt} = 2((1 + O(d^{-1} + d^{2(\gamma_{\sigma^2}+\gamma_w-1)}))a_i d - s_i)s_i(1 + O(d^{2(\gamma_{\sigma^2}+\gamma_w-1)}))$$

Let $T_5$ be the first time after $T_4$ when $|s_i - a_i d| \leq O(d^{-1} + d^{2(\gamma_{\sigma^2}+\gamma_w-1)})\log d$. Then

$$T_5 - T_4 \leq -\frac{\max(-1, 2(\gamma_{\sigma^2} + \gamma_w - 1))}{2a_K}d\log d + O(d\log\log d).$$

We conclude that at time

$$T^* = \left(\frac{1 - \gamma_{\sigma^2} - \gamma_w}{a_K} + \frac{2\max(1, 2(-\gamma_{\sigma^2} - \gamma_w + 1))}{c(a_1, \ldots, a_K)} + \frac{\max(1, 2(-\gamma_{\sigma^2} - \gamma_w + 1))}{2a_K}\right) d\log d$$
$$+ O(d\log\log d)$$

we have

$$x(T^*) \leq O(d^{\max(-1, 2(\gamma_{\sigma^2}+\gamma_w-1))}),$$
$$A_{\theta(T^*)} \in P_2(C, \max\{1, 2(1 - \gamma_{\sigma^2} - \gamma_w)\}),$$

and

$$|s_i(T^*) - a_i d| \leq O(d^{-1} + d^{2(\gamma_{\sigma^2}+\gamma_w-1)})\log d$$

### E.4 Analysis of Testing Error

In section E.3 we proved that $A_{\theta(t)}$ is almost aligned with $A^*$ throughout the training. With the alignment in hand, we are ready to give a time to stop training and the testing error at the end of training.

**Theorem E.5.** *Assume that $A_{\theta(t)}$ follows the gradient flow dynamics. At time*

$$T^* = \left( \frac{1 - \gamma_{\sigma^2} - \gamma_w}{a_K} + \frac{2max(1, 2(-\gamma_{\sigma^2} - \gamma_w + 1))}{c(a_1, \ldots, a_K)} + \frac{max(1, 2(-\gamma_{\sigma^2} - \gamma_w + 1))}{2a_K} \right) d \log d + O(d \log \log d),$$

*the testing error*

$$\|A_{\theta(T^*)} - A^*\|_F^2 \leq O(\sigma^4 w d^2) + O(\sigma^4 w^2 \log^2 d) + O(d^{\frac{3}{2}}) + O(\sigma^2 w d)$$

*Proof.* Let $P_K$ be the projection to the largest $K$ singular values. Then

$$\|A_{\theta(t)} - A^*\|_F^2 \leq \|P_K A_{\theta(t)} - A^*\|_F^2 + \|(I - P_K)A_{\theta(t)}\|_F^2.$$

Let $s_1, \ldots, s_d$ be the singular values of $A_{\theta(t)}$. Then $\|(I - P_K)A_{\theta(t)}\|_F^2 = \sum_{p \geq K+1} s_p^2$. The derivative of $s_1, \ldots, s_d$ reads

$$d^2 \frac{dS}{dt} = I \odot \left( U^T(A^* + E)V\sqrt{S^2 + \sigma^4 w^2} + \sqrt{S^2 + \sigma^4 w^2}U^T(A^* + E)V - 2S\sqrt{S^2 + \sigma^4 w^2} \right)$$
$$+ I \odot \left( U^T(A^* + E)VO(\sigma^2 w_1) + O(\sigma^2 w_1)U^T(A^* + E)V - 2SO(\sigma^2 w_1) \right)$$

If $p \geq K + 1$, then $s_p \leq C'\sigma^2 w$ for some constant $C'$, and

$$d^2 \left| \frac{ds_p}{dt} \right| \leq \left| (U^T A^* V)_{pp} + (U^T E V)_{pp} \right| O(\sigma^2 w).$$

$$d^2 \frac{d}{dt} \sum_{p \geq K+1} s_p^2 \leq \sum_p s_p \left( \sum_{q \leq K} U_{qp} V_{qp} A^*(qq) + (U^T A^* V)_{pp} \right) O(\sigma^2 w)$$

$$\leq \sum_p s_p \left( \sum_{q \leq K} |U_{qp}||V_{qp}|O(d) + O(\sqrt{d}) \right) O(\sigma^2 w)$$

$$\leq \sum_{q \leq K} \left( \sum_p s_p^2 \right)^{\frac{1}{2}} \left( \sum_p |U_{qp}|^2 \right)^{\frac{1}{2}} \max_p |V_{qp}|O(\sigma^2 w d) + \left( \sum_p s_p^2 \right)^{\frac{1}{2}} O(\sigma^2 w d)$$

$$\leq \left( \sum_p s_p^2 \right)^{\frac{1}{2}} \left( O(d^{1-\gamma}\sigma^2 w) + O(\sigma^2 w d) \right)$$

We conclude that

$$d^2 \frac{d}{dt} \|(I - P_K)A_{\theta(t)}\|_F \leq O(\sigma^2 w d),$$

which implies that

$$\|(I - P_K)A_{\theta(t)}\|_F \leq \sqrt{d}O(\sigma^2\sqrt{w}d) + O(\sigma^2 w d \frac{\log d}{d}) = O(\sigma^2\sqrt{w}d + \sigma^2 w \log d)$$

$$\|(I - P_K)A_{\theta(t)}\|_F^2 \leq O(\sigma^4 w d^2 + \sigma^4 w^2 \log^2 d)$$

Next we estimate $\|P_K A_{\theta(t)} - A^*\|_F^2$. Notice that $\|P_K A_{\theta(t)} - A^*\|_F^2 = \sum_{i,j} \|P_K A_{\theta(t)}(i,j) - A^*(i,j)\|_F^2$. If $i \neq j$, then $A^*(i,j) = 0$, and

$$\|P_K A_{\theta(t)}(i,j)\|_F^2 \leq \sum_{k:signal, k \neq i} \|U(i,k)\|_F^2 \|S(k,k)V(j,k)^T\|_F^2$$
$$+ \|U(i,j)S(j,j)\|_F^2 + \|V(j,k)^T\|_F^2$$
$$\leq O(xd^2)$$

If $i = j = m + 1$, we also have $A^*(m+1, m+1) = 0$, and

$$\|P_K A_{\theta(t)}(m+1, m+1)\|_F^2 \leq \sum_{k:\text{signal}} O(d^2)\|U(m+1,k)\|_F^2 + \|V(m+1,k)\|_F^2 \leq O(d^2 x)$$

Now assume that $i = j$ are both signals. Then

$$\|P_K A_{\theta(t)}(i,i) - A^*(i,i)\|_F^2 \leq \|U(i,i)(S(i,i) - S^*(i,i))V^T(i,i)\|_F^2 + O(d^2)\|U(i,i)V(i,i)^T - I\|_F^2$$

$$\leq O(d^{-1} + d^{2(\gamma_{\sigma^2} + \gamma_w - 1)})\log d + O(d^2\sqrt{x})$$

Since there are only finitely many blocks in total, we conclude that at time $T^*$,

$$\|A_{\theta(T^*)} - A^*\|_F^2 \leq O(\sigma^4 w d^2 + \sigma^4 w^2 \log^2 d + d^2 O(d^{\max(-\frac{1}{2},(\gamma_{\sigma^2}+\gamma_w-1))})) + O(d^{-1} + d^{2(\gamma_{\sigma^2}+\gamma_w-1)})\log d$$

$$\|A_{\theta(T^*)} - A^*\|_F^2 \leq O(\sigma^4 w d^2) + O(\sigma^4 w^2 \log^2 d) + O(d^{\frac{3}{2}}) + O(\sigma^2 w d) \tag{41}$$

$\square$

# F    Gradient Descent Dynamics and Proof of Theorem 2

In this section we prove that gradient descent dynamics of $A_{\theta(t)}$ is well-approximated by gradient flow dynamics of $A_{\theta(t)}$.

## F.1    Gradient Descent vs Gradient Flow

To study the dynamics of $A_t$ under gradient flow, we show that if the learning rate is small enough, then the gradient flow dynamics will be close to the gradient descent dynamics.

**Lemma F.1.** *Assume that $A$ is a matrix (not necessarily square matrix). $F$ is a function: $\mathbb{R}^{\dim A} \to \mathbb{R}^{\dim A}$. The norm $\|\cdot\|$ satisfies $\|AB\| \leq \|A\|\|B\|$ for all $A$ and $B$. In particular, operator norm and Frobenius norm satisfies this property. Assume that $\sup_A \|F(A)\| \leq C_0$ and $\|\nabla F\| \leq C_1$ for some constant. Consider gradient flow dynamics and gradient descent dynamics.*

$$\text{Gradient Flow: } \frac{dA_f}{dt} = F(A_f)$$

$$\text{Gradient Descent: } \frac{A_d((k+1)\eta) - A_d(k\eta)}{\eta} = F(A_d(k\eta))$$

*Assume that $A_f(0) = A_d(0)$. Then*

$$\|A_f - A_d\|(k\eta) \leq ((1 + \eta C_1)^{k-1} - 1)\frac{1}{2}\eta C_0$$

*Proof.* Notice that we have

$$A_f((k+1)\eta) - A_f(k\eta) = \int_0^\eta F(A_f(k\eta + t))dt$$

$$(A_f - A_d)((k+1)\eta) - ((A_f - A_d)(k\eta)) = \int_0^\eta F(A_f(k\eta + t)) - F(A_d(k\eta))dt$$

$$\int_0^\eta F(A_f(k\eta + t)) - F(A_d(k\eta))dt = \int_0^\eta F(A_f(k\eta + t)) - F(A_f(k\eta))dt$$
$$+ \eta(F(A_f(k\eta)) - F(A_d(k\eta)))$$

Let $G(t) = \int_0^t F(A_f(k\eta + s))ds$. Then

$$\int_0^\eta F(A_f(k\eta + t)) - F(A_f(k\eta))dt = G(\eta) - G(0) - \eta G'(0) - \frac{1}{2}\eta^2 G''(\xi)$$

$$= \frac{1}{2}\eta^2 \frac{d}{dt}|_{t=\xi} F(A_f(k\eta + t))$$

$$= \frac{1}{2}\eta^2 \frac{\partial F}{\partial A}(A_f(k\eta + \xi))\frac{d}{dt}|_{t=\xi} A_f(k\eta + t)$$

$$= \frac{1}{2}\eta^2 \frac{\partial F}{\partial A}(A_f(k\eta + \xi))F(A_f(k\eta + \xi))$$

$$\left\| \int_0^\eta F(A_f(k\eta + t)) - F(A_f(k\eta))dt \right\| \leq \frac{1}{2}\eta^2 C_0 C_1$$

$$\eta\|(F(A_f(k\eta)) - F(A_d(k\eta)))\| \leq \eta\|\nabla F\|\|A_f(k\eta) - A_d(k\eta)\| \leq \eta C_1\|A_f(k\eta) - A_d(k\eta)\|$$

We conclude that

$$\|A_f((k+1)\eta) - A_d((k+1)\eta)\| \leq (1 + \eta C_1)\|A_f(k\eta) - A_d(k\eta)\| + \frac{1}{2}\eta^2 C_0 C_1$$

$$\|A_f - A_d\|((k+1)\eta) + \frac{1}{2}\eta C_0 \leq (1 + \eta C_1)(\|A_f - A_d\|(k\eta) + \frac{1}{2}\eta C_0)$$

$$\leq (1 + \eta C_1)^k \frac{1}{2}\eta C_0$$

$$\|A_f - A_d\|(k\eta) \leq ((1 + \eta C_1)^{k-1} - 1)\frac{1}{2}\eta C_0$$

$\square$

To apply the lemma above we need to prove that $\|A_{\theta(t)}\|_F^2 \leq O(d^2)$ throughout the training.

**Lemma F.2.** *For both lazy and active regime, we always have $\|A_{\theta(t)}\|_F^2 \leq 10(\|A^*\|_F^2 + \|E\|_F^2)$ throughout the training.*

*Proof.* The gradient descent dynamics of $A_{\theta(t)}$ is given by

$$A_{\theta(t+1)} - A_{\theta(t)} = \eta d^{-2}(A^* + E - A_{\theta(t)})C_1 + C_2(A^* + E - A_{\theta(t)}).$$

$$tr((A^* + E - A_{\theta(t+1)})^T(A^* + E - A_{\theta(t+1)}) - (A^* + E - A_{\theta(t)})^T(A^* + E - A_{\theta(t)}))$$
$$= -2tr((A^* + E - A_{\theta(t)}))^T(A_{\theta(t+1)} - A_{\theta(t)})) + \|A_{\theta(t+1)} - A_{\theta(t)}\|_F^2$$
$$\leq -2\eta d^{-2}tr((A^* + E - A_{\theta(t)})^T(C_1 + C_2)(A^* + E - A_{\theta(t)})) + \eta^2 O(d^{-2}\|C_1\|_{op}^2)$$

From theorem 2, we know that $C_1 + C_2 \geq \frac{1}{3}\sigma^2 wI$. Therefore $tr((A^* + E - A_{\theta(t)})^T(C_1 + C_2)(A^* + E - A_{\theta(t)})) \geq \|A^* + E - A_{\theta(t)}\|_F^2 \frac{1}{3}\sigma^2 w \geq cd^2\sigma^2 w$ for some constant $c$. Therefore for the lazy regime, we always have $\|A^* + E - A_{\theta(t)}\|_F^2(t+1) \leq \|A^* + E - A_{\theta(t)}\|_F^2(t)$ if $5(\|A^*\|_F^2 + \|E\|_F^2) \leq \|A_{\theta(t)}\|_F^2 \leq 7(\|A^*\|_F^2 + \|E\|_F^2)$, which implies that $\|A_{\theta(t)}\|_F^2 \leq 10(\|A^*\|_F^2 + \|E\|_F^2)$ for all time. For the active regime, we have

$$\|A^* + E - A_{\theta(t)}\|_F^2(t+1) - \|A^* + E - A_{\theta(t)}\|_F^2(t) \leq \eta^2 O(1).$$

Since the training has at most $O(\frac{T^*}{\eta})$ steps, we see that $\forall t, \|A^* + E - A_{\theta(t)}\|_F^2 \leq 2(\|A^*\|_F^2 + \|E\|_F^2) + O(\eta T^*) \leq 10(\|A^*\|_F^2 + \|E\|_F^2)$. $\square$

*Proof of main theorem.* We first consider the active regime. It suffices to consider the error from considering gradient descent, rather than gradient flow. To apply lemma F.1, it is more convenient to consider the dynamics for $W_1$ and $W_2$. By lemma G.2, $\|W_1^T W_1\|_F^2 + \|W_2 W_2^T\|_F^2 \leq O(d^2)$ and $\|W_1\|_{op} + \|W_2\|_{op} \leq O(\sqrt{d})$. The gradient flow dynamics of $[W_1, W_2^T]$ is given by

$$\frac{d}{dt}[W_1, W_2^T] = d^{-2}[W_2^T(A^* - W_2 W_1), W_1(A^* - W_2 W_1)^T].$$

Let $F([W_1, W_2^T]) = d^{-2}[W_2^T(A^* - W_2 W_1), W_1(A^* - W_2 W_1)^T]$. Then $\sup_t \|F(W_1, W_2^T)\|_F \leq O(d^{-\frac{1}{2}})$. Computing the differential of $F$, we obtain that

$$dF([W_1, W_2^T]) = d^{-2}[-dW_2^T(A^* - W_2 W_1) + W_2^T(-dW_2 W_1 - W_2 dW_1),$$
$$- dW_1(A^* - W_2 W_1)^T + W_1(-dW_1^T W_2 - W_1^T dW_2)]$$

and therefore $\|\nabla F\|_F \leq O(d^{-1})$. In the active regime, the total number of training steps is $\eta^{-1}O(d\log d)$ By lemma F.1,

$$\|W_1^{flow} - W_1^{descent}\|_F(T^*) \leq O(\eta d^{-\frac{1}{2}})((1 + \eta O(d^{-1}))^{\frac{O(d\log d)}{\eta}} - 1) = O(\eta d^{-\frac{1}{2}}\log d)$$

and the same holds true for $W_2$. We conclude that

$$\|W_2^{flow}W_1^{flow} - W_2^{descent}W_1^{descent}\|_F$$
$$\leq (\|W_2^{descent}\|_{op} + \|W_1^{descent}\|_{op})O(\eta d^{-\frac{1}{2}}\log d)$$
$$= O(\eta \log d)$$

$$\|A_\theta^{flow} - A_\theta^{descent}\|_F^2 \leq O(\eta^2 \log^2 d).$$

In the lazy regime, from strong bound we have

$$W_1^T W_1 = (1 + O(\sqrt{\frac{d}{w}}))\sqrt{\sigma^4 w^2 I + A^T A}.$$

Therefore $\|W_1\|_{op} \leq O(\sigma\sqrt{w})$, $\sup_t \|F([W_1, W_2^T])\|_F \leq d^{-2}(\|W_1\|_{op} + \|W_2\|_{op})O(d) = O(d^{-1}\sigma\sqrt{w})$. Similarly, $\|\nabla F\|_F \leq d^{-2}O(\|A^* - W_2 W_1\|_F + \|W_2\|_F \|W_1\|_F) = d^{-2}O(\sigma^2 w)$. By lemma F.1, at time $\frac{100d^2 \log d}{\sigma^2 w}$,

$$\|W_1^{flow} - W_1^{descent}\|_F(\frac{100d^2 \log d}{\sigma^2 w}) \leq O(\eta d^{-1}\sigma\sqrt{w})((1+\eta O(d^{-2}\sigma^2 w))^{\frac{100d^2 \log d}{\sigma^2 w \eta}} - 1) = O(\eta d^{-1}\sigma\sqrt{w}\log d).$$

We conclude that

$$\|W_2^{flow}W_1^{flow} - W_2^{descent}W_1^{descent}\|_F$$
$$\leq (\|W_2^{descent}\|_{op} + \|W_1^{descent}\|_{op})O(\eta d^{-1}\sigma\sqrt{w}\log d)$$
$$= O(\eta d^{-1}\sigma^2 w \log d).$$

$$\|W_2^{flow}W_1^{flow} - W_2^{descent}W_1^{descent}\|_F \leq O(\eta^2 d^{-2}\sigma^2 w \log^2 d).$$

Recall that in lemma D.1 we proved that

$$\|A_{\theta(t)}^{flow} - B_t\|_F^2 = o(d^2),$$

and at time $100\frac{d^2 \log d}{\sigma^2 w}$, $\|B_t - A^* - E\|_F^2 \leq d^{-50}$, which implies that $\|A_\theta^{descent} - A^* - E\|_F^2 \leq o(d^2)$. Before this time, we have $\|B_t - A^*\|_F^2 \geq \frac{1}{3}\min(\|A^*\|_F^2, \|E\|_F^2)$. After this time, we have

$$tr((A^* + E - A_{\theta(t+1)}^{descent})^T(A^* + E - A_{\theta(t+1)}^{descent}) - (A^* + E - A_{\theta(t)}^{descent})^T(A^* + E - A_{\theta(t)}^{descent}))$$
$$= -2tr((A^* + E - A_{\theta(t)}^{descent}))^T(A_{\theta(t+1)}^{descent} - A_{\theta(t)}^{descent})) + \|A_{\theta(t+1)}^{descent} - A_{\theta(t)}^{descent}\|_F^2$$
$$\leq -2\eta d^{-2}tr((A^* + E - A_{\theta(t)}^{descent})^T(C_1 + C_2)(A^* + E - A_{\theta(t)}^{descent}))$$
$$+ \eta^2 tr((A^* + E - A_{\theta(t)}^{descent})^T(10\sigma^4 w^2 I)(A^* + E - A_{\theta(t)}^{descent}))$$
$$\leq -(2\eta d^{-2}\sigma^2 - 10\eta^2 \sigma^4 w^2)\|A^* + E - A_{\theta(t)}^{descent}\|_F^2$$

Therefore $\|A^* + E - A_{\theta(t)}^{descent}\|_F^2$ is decreasing and therefore $\|A_\theta^{descent} - A^*\|_F^2 \geq \frac{1}{3}\min(\|A^*\|_F^2, \|E\|_F^2)$ for all time.

$\square$

# G Experimental setup

We now describe the experimental setup for the experiments shown in Figures 1 and 2. For all the experiments, we used the losses

$$\mathcal{L}_{\text{train}}(\theta) = \frac{1}{d^2}\|A_\theta - (A^* + E)\|_F^2; \quad \mathcal{L}_{\text{test}}(\theta) = \frac{1}{d^2}\|A_\theta - A^*\|_F^2$$

where $E$ has i.i.d. $\mathcal{N}(0,1)$ entries, $A^* = K^{-1/2}\sum_{i=1}^K u_i v_i^T$ with $u_i, v_i \sim \mathcal{N}(0, \text{Id}_d)$ Gaussian vectors in $\mathbb{R}^d$. This means that $\text{Rank}A^* = K$. The factor $K^{-1/2}$ ensures that $\|A^*\|_F = \Theta(d)$.

We then either run the self-consistent dynamics (equation (1)) or gradient descent (equation (2)). Following Theorem 2, we take a learning rate $\eta = \frac{d^2}{cw\sigma^2}$ for $\gamma_{\sigma^2} + \gamma_2 > 1$, and $\eta = \frac{d^2}{c\|A^\star\|_{\mathrm{op}}}$ otherwise, where $c$ is usually 50 but can be taken to be 2 or 5 for faster convergence at the cost of more unstable training.

For the experiments in Figure 1, we took $d = 500$ and $K = 5$. For the experiments in Figure 2, we took $d = 200$ and $K = 5$. For making the contour plot, we took a grid with 35 points for $\gamma_{\sigma^2} \in [-3.0, 0.0]$ and 35 points for $\gamma_w \in [0, 2.8]$. For each of the $35^2$ pair of values for $(\gamma_{\sigma^2}, \gamma_w)$, we ran gradient descent (and for the lower right plot the self-consistent dynamics too) until the train error converged. For all the runs, we took the same realizations of $A^\star$ and $E$.

All the experiments were implemented in PyTorch [40]. Experiments took 12 hours of compute, using two GeForce RTX 2080 Ti (11GB memory) and two TITAN V (12GB memory).

