# OpenReview forum: "Mixed Dynamics In Linear Networks: Unifying the Lazy and Active Regimes"
_NeurIPS.cc/2024/Conference — NeurIPS 2024 poster_

### Official Review · Reviewer_pxfL · 2024-07-09

**Soundness:** 4
**Presentation:** 3
**Contribution:** 3
**Rating:** 7
**Confidence:** 3

**Summary:**

In this paper, the authors study the training dynamics of a two-layer linear network $A = W_1 W_2$ and show that for a wide range of network width $w$ and initialization scale $\sigma^2$, the dynamics of $A$ can be approximated by the self-consistent equation:
$\dot{A} \approx -\sqrt{ A A^T + \sigma^4 w I } \nabla C(A) - \nabla C(A) \sqrt{ A^T A + \sigma^4 w I }$,
which can be viewed as a mixture of the lazy and balanced dynamics.
With this observation, the authors then analyze the behavior of gradient descent with different choices of $w$ and $\sigma^2$ for the task of recovering a low-rank matrix from noisy observations. In particular, they identify two possible regimes for $w$ and $\sigma^2$: the pure lazy regime, where the network fails to recover the ground-truth matrix, and a mixed/active regime, where the network first aligns with the ground-truth following the lazy dynamics and then switches to the balanced (and aligned) dynamics and eventually recovers the ground-truth.

**Strengths:**

This paper unifies the lazy and balanced regimes of the training dynamics of two-layer linear networks. The equation (cf. (1)) obtained by the authors that approximately characterize the dynamics is surprisingly simple and suggests a way to classify the training behaviors that are more fine-grained than the lazy vs balanced regimes. The proof of Theorem 1 (the theorem on the validity of the approximation (1)) also looks interesting. Instead of controlling the error growth rate like in the proof of many similar results, it relies on the fact that $W_1W_1^T - W_2^T W_2$ is invariant under GF to obtain an equation and the stability of the solutions to that equation.

With the new equation, the authors also prove a global convergence result for balanced initialization without assuming alignment at initialization. This is a novel result and demonstrates the usefulness of the new characterization, as it leverages the initial short lazy regime to obtain an approximately aligned state.

**Weaknesses:**

1. The presentation of the paper can still be improved. In particular, some parts of the paper seem to be rushed (the appendix in particular) and contain many typos (e.g. the second equation in line 212, line 761, and line 773).

2. As mentioned by the authors in Section~2.1, in the low-rank matrix recovery setting, the width of the network is assumed to be much larger than the ambient dimension (instead of the rank). What will happen if the network width is much larger than the rank but much smaller than the ambient dimension? It seems that the network will be directly in the active regime and we can no longer rely on the initial lazy stage to align the network. Is this true in theory/practice?

**Questions:**

1. Could you provide some intuition on the proof of Theorem~1? It seems to me to be some clever algebraic manipulations that cannot be easily explained intuitively.


2. See item 2 of Weakness.

**Limitations:**

See Weakness.

---

> ### Author Rebuttal · Authors · 2024-08-06
>
> Thanks for the thoughtful review. Regarding the weaknesses you mention:
>
> 1. We will improve the readability of the proofs, thanks to the error/typos
> that you and the other reviewers have found.
>
> 2. We agree that this intermediate width regime is of particular interest,
> and it is probably the regime where one wants to be in practice, and
> we have thought of it but it seems that it would require different
> techniques (in particular the invariant approach might not work anymore).
> While your intuition that the dynamics would be purely balanced could
> be correct, there is another possible interpretation: when the task
> has a low rank structure, then only the dynamics along the low-dimensional
> span of the true matrix matter, the parameters orthogonal to remain
> essentially constant, at least until the optimal early stopping time,
> thus the dynamics could possibly be approximated by a smaller network
> with the same hidden layer size but with input and output of size
> equal to the rank of the true matrix $A^{*}$. We hope to be able
> to extend our analysis to this setup to answer which intuition is
> correct.
>
> Regarding your question:
>
> 1. We will improve the proof and add a intuition section in the Appendix to explain the strategy and how we rule out the other solutions. Depending on how much room we have, we will also explain this in the main. Another to understand Theorem 1
> is to show an equivalence to another dynamic as described in our answer
> to Reviewer r8Jt, we will also add this simple derivation to help with
> the intuition.

---

> > ### Comment · Reviewer_pxfL · 2024-08-08
> >
> > Thank you for your clarifications. I will maintain my score.

---

### Official Review · Reviewer_mU2H · 2024-07-11

**Soundness:** 2
**Presentation:** 3
**Contribution:** 2
**Rating:** 5
**Confidence:** 4

**Summary:**

This paper derives a formula for the training dynamics of two-layer linear networks, encompassing the lazy regime, the active regime, and the mixed regime.

**Strengths:**

- In constract to previous works, the authors reveal the existence of mixed dynamics for training two-layer linear networks, which combine the advantages of the lazy regime and the active regime: it converges from any random initialization, and has a low rank bias.

- The authors prove an almost complete phase diagram of training behavior  on the task of recovering a low-rank matrix from noisy observations.

**Weaknesses:**

The main findings heavily rely on prior insights into lazy training and feature learning. Moreover, the focus on two-layer linear networks, which are simple and lack nonlinearity, limits the broader impact of this paper.

**Questions:**

- Do the mixed dynamics have a stronger or weaker implicit bias compared to the active regime (such as the low-rank bias or sparsity bias)? Which regime predominates in practical scenarios?

- Can the theory recover other non-trivial training dynamics of linear networks, such as the saddle-to-saddle dynamics [1]?

- How can the main insights from this study be generalized to nonlinear models, such as two-layer ReLU nets?

- Minor: There appears to be a missing '-' in the equation between line 55 and line 56.

[1] Pesme \& Flammarion. Saddle-to-Saddle Dynamics in Diagonal Linear Networks. (NeurIPS 2023)

**Limitations:**

The analysis in this study focus on 2-layer linear networks, which is limited.

---

> ### Author Rebuttal · Authors · 2024-08-06
>
> Thanks for the thoughtful review. Regarding your questions:
>
> - Note that we view the mixed dynamics as part of the active regime,
> what we show is that the short lazy dynamic period that always appears
> at the beginning plays an important role, so it is useful to think
> of the active regime as a mix of lazy and active. The balanced dynamics
> that start after the short lazy period then leads to a low-rank bias.
>
> - We are describing the saddle-to-saddle dynamics actually, and improving
> significantly on prior results, since we prove it in the fully-connected,
> non-diagonal case which is much harder. Furthermore we can determine
> exactly how small the variance $\sigma^{2}$ needs to be to get the
> saddle-to-saddle dynamics, in comparison to previous works that required
> an infinitely small initialization (https://arxiv.org/abs/2012.09839
> or https://arxiv.org/abs/2106.15933) and could not really handle the
> second saddle making it difficult to determine when one would leave
> each subsequent saddle. We solve all of these issues. We will add a
> discussion of the relation to the Saddle-to-Saddle dynamics.
>
> - There are a few high level ideas that could translate to nonlinear
> nets: (1) there could exist mixed regimes in nonlinear networks,
> where e.g. some neurons are active while others are still lazy in
> some sense; (2) the transition from lazy to active could shift depending
> on the sparsity of the task, which could explain why we are not yet
> able to fully describe the extent of the lazy regime as a function
> of initialization variance, width, and number of datapoints and how
> it depends on the task at hand.

---

> > ### Comment · Reviewer_mU2H · 2024-08-09
> >
> > Thanks for the authors' detailed responses to my questions. The authors have addressed my concerns, and I have raised my score.

---

### Official Review · Reviewer_oPFX · 2024-07-12

**Soundness:** 2
**Presentation:** 2
**Contribution:** 3
**Rating:** 6
**Confidence:** 3

**Summary:**

This paper consider the GD dynamics for two layer linear network. The authors introduce an approximated dynamics which interpolate between the lazy and the balanced regime. The authors also showed the phase diagram based on the above dynamics for low-rank matrix factorization problem.

**Strengths:**

1. This paper provides an approximated dynamics GD dynamics for two-layer linear network, including different initialization scheme. I believe this is novel in the literature and is an interesting results.

**Weaknesses:**

1.  Regarding the (pure) lazy training part, it is said in Line 124 "When $w$ is very large, we end up in the lazy regime where the parameters move enough up to a time $t$ to change $A\_{θ(t)}$ , but not enough to change $C_1 , C_2$" . Could you explain more on this or point out some references? In particular why $C_i(t) \approx C_i(0)$ for any $t$?

2. The statement of Theorem 1 needs to be written more clearly: (1) what is $\delta$ in the statement? Do you mean that the upper bounds hold for any $0<\delta<1$? Then how do the upper bounds depend on $\delta$? (2) what is $C_1$ in the RHS of the bound. From the proof I think you mean $C_1(t)$ not $C_1(0)$?Then how does $C_1(t)$ depend on $t$, in particular, does it blow up in $t$?

3. I also have a few doubts on the utility of Theorem 1: it is not clear to me whether the RHS of the upper bound in Theorem 1 vanishes in $d$ ? In particular, consider the settings in section 2.1 where $\sigma^2 = d^{\gamma\_{\sigma^2}},w = d^{\gamma\_{w}}$, then the first term  $\mathcal{O}( \sigma^2 w)$ is not vanishing in $d$ unless $\gamma\_{\sigma^2} +  \gamma\_{w}<0$. For the second term $ O( \sqrt{d/w} ||C_1||\_{op} )$, it is not clear to me how $ ||C_1(t)||\_{op}$ depends on $d$ for any $t$. Thus, could you address more on this point?

In conclusion, so far the utility of the main theorems is not clear to me, I will raise my scores if my questions are addressed.

**Questions:**

Please refer to the strengths and weaknesses part.

**Minor points and typos:**
1. Equation (1) and the equation below Line 204 seems to be different in a $\eta$ factor.
2. In Line 284 should be $2 \gamma_{\sigma^2} + \gamma_w < 0$.
3. Line 939, where is Lemma G.2?

**Limitations:**

No potential negative societal impact

---

> ### Author Rebuttal · Authors · 2024-08-06
>
> Thanks for the thoughtful review. Regarding the weaknesses you raise:
>
> 1. Proving this is non-trivial and would require some work to be proven
> directly, the intuition is that one needs to take a learning rate
> of order $\sigma^{-1}$ to get finite size updates to $A_{\theta}$,
> but the change to the weight matrices $W_{1},W_{2}$ is then of order
> $1$ which is small relative to their size at initialization (of order
> $\sigma$). Note however that this fact also follows from our first
> theorem, if $\sigma$ is sufficiently large then $C_{1}=\sqrt{A_{\theta(t)}^{T}A_{\theta(t)}+w^{2}\sigma^{4}I}\approx\sigma^{2}wI$
> for all times $t$.
>
> 2. (1) Yes any $\delta$ can be chosen, actually we will remove $\delta$ from the Theorem and simply say "with high probability"
> (2) It should indeed be $C_{1}(t)$. Note
> that $C_{1}$ might indeed blow up in time, but we are mainly interested
> in having a small error relative to the size of $C_{1}(t)$ (the learning
> rate $\eta$ has to be chosen of order $\left\Vert C_{1}\right\Vert _{op}^{-1}$
> so things only need to be small in comparison to $\\left\\Vert C\_{1}\\right\\Vert\_{op}$).
> The fact that in Theorem 2 we are able to control the dynamics up
> until convergence using Theorem 1 shows that the approximation of
> Theorem 1 is good enough to be used in a practical setting. We will
> add a discussion of these aspects after Theorem 1, because we agree
> that it can be difficult to determine when such an approximation is
> "good enough" since everything can vary
> significantly in size as $d\to\infty$ and throughout training.
>
> 3. Again the RHS does not always go to zero in $d$ but it always
> becomes infinitely smaller than $\left\Vert C_{1}(t)\right\Vert _{op}$
> which is sufficient.

---

> > ### Comment · Reviewer_oPFX · 2024-08-12
> >
> > Thank you for your detailed response. Now I have a better understanding of the theoretical contribution of this work, and I believe it is an interesting result. Thus I raise my score to 6.

---

### Official Review · Reviewer_r8Jt · 2024-07-13

**Soundness:** 3
**Presentation:** 3
**Contribution:** 3
**Rating:** 6
**Confidence:** 3

**Summary:**

The paper studies gradient descent dynamics in two-layer linear networks. It is shown that in the wide-hidden-layer regime with standard random initialization there is a simple self-consistent differential equation describing the network output (Theorem 1). This equation generalizes both lazy training regime and the standard solvable balanced regime. As an application, a phase diagram of the model with respect to different scalings of the hidden layer size and the weight variance is discussed (section 2.1).

**Strengths:**

The paper is generally nicely and thoughtfully written.

The main theoretical result described in the paper (the self-consistent evolution equation) is interesting and apparently original. Thanks to the simplicity of this equation, it is likely to be useful in future research.

This paper uncovers a simple and natural scenario in which lazy training transitions into an active training in an analytically tractable fashion. This scenario can help understand feature learning in more complex models.

**Weaknesses:**

I find the exposition of the main result (Theorem 1) and surrounding ideas not very clear.
The provided sketch of proof is not very convincing, since the derived cubic equations can generally have solutions $C_1,C_2$ other than those indicated. I also couldn't get through the proof in the appendix, particularly Proposition C.4. Some typos there:
line 689: "*Let* $L = ||A^*-A ||^2$ *be the loss function*"  - not used in the proof
line 761: "*Recall that $\tfrac{dw}{dt}=$*" - unfinished sentence
line 773: "*To show that $∥v_i^TdC_1∥$ and $∥u_i^T dC_2∥$*" - unfinished statement

**Questions:**

It seems that a linear network combining lazy and active regime can actually be constructed more directly, within the class of balanced models, by simply considering a non-homogeneous balance condition $W_1W_1^T=W_2^TW_2-b^2I$. This condition is realized, in particular, when the initial $W_1(t=0)=0$ while the initial $W_2$ is isometric up to rescaling: $W_2^TW_2(t=0)=b^2I$. The balance condition is again invariant and, arguing as in the paper, we can obtain the closed-form dynamics for $A=W_2W_1$. Namely,  by multiplying the invariant by the matrices $W_1, W_2$ or their transposes, we get the equations  $C_1^2+b^2C_1-A^TA=0$ and $C_2^2-b^2C_2-AA^T=0$. Finding the roots leads to the self-consistent GF equation $$\tfrac{dA}{dt}=-\tfrac{\eta}{2}[(\sqrt{4AA^T+b^4}+b^2)\nabla C+\nabla C(\sqrt{4A^TA+b^4}-b^2)]=-\tfrac{\eta}{2}[\sqrt{4AA^T+b^4}\nabla C+\nabla C\sqrt{4A^TA+b^4}].$$ This dynamics also starts in a lazy regime, because initially $W_1=0$ so that $W_2(t)\approx const$ and learning is linear and occurs only through $W_1$. As deviation of $W_1$ from 0 increases, the dynamics becomes active.

My impression from the proof of Theorem 1 is that it is essentially a reduction of the wide network model to the algebraically solvable "balanced-type" model, but with two channels corresponding to the two approximally orthogonal projectors $P_1, P_2$ appearing in Theorem 1. By restricting to the corresponding subspaces, the matrices $W_1, W_2, A$ are decomposed into two components, say $W_1=Y_1+Z_1, W_2=Y_2+Z_2$ and $A=A_Y+A_Z\equiv Y_2Y_1+Z_2Z_1$. The initial conditions in one channel are like in the example above, $Y_1(t=0)\approx 0, Y_2^TY_2(t=0)=\sigma^2 wI$, while in the other channel they are reversed, $Z_2(t=0)\approx 0, Z_1Z_1^T(t=0)=\sigma^2 wI$. However, both channels are described by the same equation with the same initial condition, so yield the same solution $A_Y=A_Z.$ Then the equation for the total $A=A_Y+A_Z$ presented in the paper can be obtained from the single-channel equation above simply by replacing $A$ by $A/2$. I think that the paper would be easier to understand and more convincing if such a two-channel model was explicitly described, and its solution (either this or performed in the paper) explained more carefully.

**Limitations:**

The model considered in the paper (a two-layer linear network with a wide hidden layer) is very simple and fairly artificial. However, the effects exposed in the paper may be relevant for more complex and realistic models.

---

> ### Author Rebuttal · Authors · 2024-08-06
>
> Thanks for the thoughtful review. Regarding the weaknesses you mention:
>
> It is indeed true that there are many solutions to the set of equations
> we obtain, and most of the work in the proof is to prove that one
> approaches the `right' solution. This part of the argument is quite
> technical and cannot be easily sketched out in the main. We will also
> fix the typos.
>
> Regarding your questions:
>
> The construction you propose is very interesting, and it can actually
> be slightly modified to recover the mixed dynamics exactly, if one
> initializes with weights $W_{1}=(
> I_{d} \\;\\; 0)^{T}$ and $W_{2}=(0 \\\;\\; I_{d})$ then the following three properties are satisfied at initialization
> and for all subsequent times
> $$
> A_{\theta}C_{1} =C_{2}A_{\theta}$$
> $$
> C_{1}^{2} =A_{\theta}^{T}A_{\theta}+I_{d}$$
> $$
> C_{2}^{2} =A_{\theta}A_{\theta}^{T}+I_{d}$$
> since it is true at initialization and the derivatives on both sides
> of the first equation match thanks to properties 2 and 3, while the
> derivatives of both sides of 2,3 match thanks to equation 1.
>
> This initialization/derivation also seems to agree with your `two
> channels' intuition.
>
> We added your derivation of mixed regime GF equation at the end of appendix C.

---

### Decision · Program_Chairs · 2024-09-25

**Decision:**

Accept (poster)

**Comment:**

This paper studies the training dynamics of linear neural networks and identifies a mixed dynamics regime that sits in the middle between the lazy & the active regime studied by prior works. The analysis shows that this regime has theoretical advantages over the previously studied cases. The review reports are generally positive about the paper and I concur with them. The authors are encouraged to incorporate the review comments in preparing the next revision.